# Multiplexed assays of human disease-relevant mutations reveal UTR dinucleotide composition as a major determinant of RNA stability

Jia-Ying Su[1,2,3,4†], Yun-Lin Wang[1†], Yu-Tung Hsieh[1], Yu-Chi Chang[1], Cheng-Han Yang[1], YoonSoon Kang[1], Yen-Tsung Huang[2], Chien-Ling Lin[1*‡]

[1]Institute of Molecular Biology, Academia Sinica, Taipei, Taiwan; [2]Institute of Statistical Science, Academia Sinica, Taipei, Taiwan; [3]Bioinformatics Program, Taiwan International Graduate Program, Academia Sinica, Taipei, Taiwan; [4]Institute of Biomedical Informatics, National Yang Ming Chiao Tung University, Taipei, Taiwan

*For correspondence:
mbcllin@gate.sinica.edu.tw

†These authors contributed equally

Present address: ‡Institute of Molecular Biology, Academia Sinica, Taipei City, Taiwan

## eLife Assessment

This **valuable** study combines massively parallel reporter assays and regression analysis to identify sequence features in untranslated regions contributing to the stability of in vitro transcribed mRNA delivered to cells. The strength of evidence presented is **solid**, although some points about half-life measurements and the relevance of identified sequence features to native transcript stability will inform future discussion surrounding the present study. Taken together, the work will be of interest to a broad swath of colleagues studying post-transcriptional gene regulation and especially to those using massively parallel reporter assays.

**Abstract** Untranslated regions (UTRs) contain crucial regulatory elements for RNA stability, translation and localization, so their integrity is indispensable for gene expression. Approximately 3.7% of genetic variants associated with diseases occur in UTRs, yet a comprehensive understanding of UTR variant functions remains limited due to inefficient experimental and computational assessment methods. To systematically evaluate the effects of UTR variants on RNA stability, we established a massively parallel reporter assay on 6555 UTR variants reported in human disease databases. We examined the RNA degradation patterns mediated by the UTR library in two cell lines, and then applied LASSO regression to model the influential regulators of RNA stability. We found that UA dinucleotides and UA-rich motifs are the most prominent destabilizing element. Gain of UA dinucleotide outlined mutant UTRs with reduced stability. Studies on endogenous transcripts indicate that high UA-dinucleotide ratios in UTRs promote RNA degradation. Conversely, elevated GC content and protein binding on UA dinucleotides protect high-UA RNA from degradation. Further analysis reveals polarized roles of UA-dinucleotide-binding proteins in RNA protection and degradation. Furthermore, the UA-dinucleotide ratio of both UTRs is a common characteristic of genes in innate immune response pathways, implying a coordinated stability regulation through UTRs at the transcriptomic level. We also demonstrate that stability-altering UTRs are associated with changes in biobank-based health indices, underscoring the importance of precise UTR regulation for wellness. Our study highlights the importance of RNA stability regulation through UTR primary sequences, paving the way for further exploration of their implications in gene networks and precision medicine.

## Introduction

### Untranslated regions of RNAs are indispensable for post-transcriptional regulation of gene expression

A mature RNA consists of three regions - a 5' untranslated region (5' UTR), the protein coding region, and a 3' untranslated region (3' UTR; *Mignone et al., 2002*). UTRs are indispensable for gene expression. For most mRNAs of higher eukaryotes, the 5' UTR is essential for ribosome entry and the 3' UTR is responsible for polyadenylation to stabilize the RNA and enhance its translation efficiency. The average length of the 5' and 3' UTRs in human is ~210 nucleotides (nt) and ~1030 nt, respectively, with mean 3' UTR length being more diverse among species and higher eukaryotes hosting longer 3' UTRs (*Pesole et al., 2001*). These structural differences among species are consistent with genomic complexity and more complex post-transcriptional regulatory mechanisms. The UTRs contain *cis*-regulatory elements that contribute to post-transcriptionally regulating gene expression, such as via protein translation control, subcellular mRNA localization, and mRNA stability upon interactions with *trans*-acting factors such as RNA-binding proteins (RBPs) and microRNAs (miRNAs) (*Barrett et al., 2012*).

### UTRs control RNA stability

RNA stability regulation serves as an mRNA quality control mechanism (e.g. nonsense-mediated mRNA decay) to gate protein production (*Schoenberg and Maquat, 2012*). Decades of research have shown that the *cis*-regulatory elements in UTRs affect mRNA stability. These sequence elements control mRNA integrity and can trigger mRNA degradation pathways by interacting with RBPs or small regulatory RNAs such as miRNAs and small interfering RNAs (siRNAs; *Garneau et al., 2007*; *Mitchell and Tollervey, 2001*). For instance, the miRNA-Argonaute complex recruits the CCR4-NOT (Carbon Catabolite Repression—Negative On TATA-less) complex to initiate deadenylation and decay (*Huntzinger and Izaurralde, 2011*). Another example is AU-rich elements (AREs, sequence elements rich in adenosine and uridine) present in the 3' UTRs of many mRNAs that provide binding sites for ARE-binding proteins that trigger the RNA degradation pathway (*Garneau et al., 2007*). Many ARE-binding proteins have been characterized to date that are involved in stability regulation of ARE-hosting mRNAs (*Barreau et al., 2005*), including Tristetrapolin (TTP), Butyrate Response Factor 1 (BRF1), Heterogeneous Nuclear Ribonucleoprotein (hnRNP) D (also known as AU-Rich Element-Binding Protein 1, AUF1), and KH-type splicing regulatory protein (KSRP), all of which destabilize mRNA, unlike ELAV Like RNA Binding Protein 1 (ELAVL1, also known as HuR) that stabilizes it (*Schoenberg and Maquat, 2012*). This regulatory mechanism necessitates physical access to the sequence elements, so structural contexts are critical (*Paschoud et al., 2006*). Complex secondary structures such as RNA G-quadruplexes (RG4s) and pseudoknots may also play a role in stability regulation. RG4s are enriched in UTRs where they regulate many post-transcriptional regulatory processes, including RNA stability (*Dumas et al., 2021*), although the detailed mechanisms remain to be elucidated. Although 3' UTR-mediated regulation gains more attention, 5' UTRs may also contribute to mRNA stability regulation. For instance, upstream open reading frames (uORF) in 5' UTRs can facilitate RNA decay in a translation-dependent manner, and RG4s in 5' UTRs reduce RNA stability in a ribosome-independent manner (*Jia et al., 2020*).

### Multiplexed reporter assays to elucidate UTR stability control

Efforts have been made to elucidate RNA stability regulation on a genome-wide scale to understand its general regulatory mechanisms. However, due to the complexity of post-transcriptional regulation, features inferred from endogenous steady-state RNA levels can be obscured by other dominant factors. For instance, studies examining cellular RNA degradation alongside transcriptional inhibition have shown that coding region length and ribosome occupancy are key determinants of RNA stability (*Neymotin et al., 2015*). Additionally, RNA stability inferred from steady-state RNA concentrations normalized against transcription rates has indicated that splice junction density is a major factor promoting RNA stability (*Agarwal and Kelley, 2022*; *Blumberg et al., 2021*). Nonetheless, these studies have not systematically decoded the influence of primary sequences on RNA stability regulation. Therefore, efforts have been made to establish massively parallel reporter assays for bulk-synthesized UTRs to unveil their governance of RNA regulation in various species. Bidirectional

promoters driving a control transcript and a green fluorescence protein (GFP) hosting various test UTRs were first used to evaluate the effect of the UTRs on fluorescence signals (*Oikonomou et al., 2014*; *Sample et al., 2019*; *Vainberg Slutskin et al., 2018*; *Wissink et al., 2016*; *Zhao et al., 2014*). Alternatively, various UTRs have been inserted into plasmids prior to cellular expression, and then the DNA and RNA levels of each construct have been compared to infer the effect of the UTRs on RNA expression (*Griesemer et al., 2021*; *Litterman et al., 2019*; *Siegel et al., 2022*). Nevertheless, because the steady-state level is a result of production and decay, these approaches cannot differentiate the effect of the UTRs on transcription, stability and, in some cases, even protein production, greatly limiting scientific interpretation. Injections or transfections of in vitro-transcribed RNAs have been used to study RNA stability. These studies have identified AREs and miRNA-binding sites as destabilizing elements and U-rich sequences as stabilizing elements in zebrafish embryos (*Rabani et al., 2017*; *Vejnar et al., 2019*; *Zhao et al., 2014*). Similar studies in human cell lines have shown that RG4 structures and A-rich sequences in 5' UTRs promote RNA decay (*Jia et al., 2020*).

## UTR mutations and disease

UTR sequence variation affects mRNA stability, translation and localization. RNA dysregulation arising from mutations in UTRs significantly and negatively affects gene regulation, which can promote phenotypical and even pathological change. According to the NHGRI-EBI GWAS Catalog, genome-wide association studies up to 2018 had uncovered that ~3.7% of disease risk/quantitative trait-associated genetic variants are located in UTRs (*MacArthur et al., 2017*; *Steri et al., 2018*). Indeed, certain studies have provided evidence that alterations to even a single nucleotide in a UTR can impact mRNA translation or transcript half-life in disease contexts. For example, a single nucleotide substitution of the 36th position in the 5' UTR of *transforming growth factor-β3* (*TGFβ3*; OMIM# 190230) or its 1723th 3' UTR position is associated with arrhythmogenic right ventricular cardiomyopathy (*Beffagna et al., 2005*). Similarly, point mutation of the *GFPT1* 3' UTR results in congenital myasthenic syndrome. GFPT1 (Glutamine-Fructose-6-Phosphate Transaminase 1) is the rate-limiting enzyme for hexosamine biosynthesis, and mutation of its 3' UTR results in a 90% reduction in protein production, potentially due to gain of a miRNA binding site (*Dusl et al., 2015*). Collectively, these studies support that the precise RNA regulation exerted by UTRs plays a critical role in controlling gene expression, with UTR mutations potentially eliciting divergent phenotypes and even severe disease.

Despite their disease relevance, a comprehensive overview of the pathogenic effects of UTR mutations is still lacking. The medical genetics community has earnestly advocated for consideration of 'UTR variants in genetic diagnostic procedures' (*Dusl et al., 2015*). Specifically, the impact of 5' and 3' UTR sequence variations on RNA stability regulation remains unclear. While there have been efforts to examine how 3' UTR variants affect steady-state RNA levels, systematic assessments of the explicit effects of disease-relevant variants in both UTRs on stability regulation are still absent. To examine potentially pathogenic UTR mutations and their links to RNA stability, we developed a massively parallel reporter assay in which human 5'/3' UTRs with disease-relevant mutations were generated in vitro, ligated with the enhanced green fluorescence protein (EGFP) coding region, and then directly transfected into human cell lines to assess their decay patterns by next-generation sequencing. Taking redundancy and interdependency of regulatory features into consideration, our approach identified that UA dinucleotides are the most influential destabilizing sequence element for RNA stability. Moreover, we found that joint regulation by 5' and 3' UTRs shapes the expression kinetics of functional gene groups. Our study unveils RNA stability determinants and delineates the importance of precise UTR control in maintaining harmonious genetic networks for human health.

## Results

### Massively parallel reporter assay (MPRA) for RNA stability

The above-described genome-wide analysis prompted the hypothesis that UTR variants that disrupt critical RNA regulatory elements may be linked to pathogenicity. Since one of the major roles of UTRs is to control RNA stability, we hypothesized that disease-relevant UTR variants may alter RNA stability. Therefore, we designed 6555 pairs of 155-nt UTR fragments centering on the variant collected from the HGMD and ClinVar disease databases, and performed time-course assays to examine the relative stability. First, we fused UTRs to the EGFP coding regions, transcribed them in vitro, and then

transfected them into human embryonic kidney cells (HEK293T) or neuroblastoma cells (SH-SY5Y), considering pervasive neurological diseases in the mutation collection. Then, we monitored the relative abundance of the reference (ref) and mutant (mt) alleles by amplicon sequencing over a time course (30, 75, 120 min for HEK293T; 20, 40, 60 min for SH-SY5Y) (*Figure 1A*). Primers targeting common reporter regions were utilized for the retrieval of UTR sequences at each time point. We estimated the decay constant and half-life ($t_{1/2}$) of each UTR according to its relative abundance over time (see Methods). We defined stability-altering variants as those for which the decay constants significantly changed relative to their ref counterparts, as determined by weighted linear regression (see Methods). We observed that variants in both 5' and 3' UTRs significantly altered RNA half-life, with slightly more variants having a negative impact on RNA stability (*Figure 1B&C*; *Supplementary file 1*).

Our results from three independent experiments are highly consistent, with a Spearman's correlation coefficient >0.93 (*Figure 1—figure supplement 1A and B*). From among 3,700 pairs of valid comparisons, 40 (1.1%) and 839 (22.8%) variants displayed significantly altered stability compared to their ref counterparts in HEK293T and SH-SY5Y cells, respectively (*Figure 1—figure supplement 1B and C*). Thus, we observed a significant effect of UTR variation on RNA stability, but their regulatory impact was strikingly divergent between the two tested cell lines (*Figure 1—figure supplement 1D*). We attribute this divergence to potential differences in translation capacity, as well as variations in the composition and concentration of RNA-binding proteins and RNases between the two cell lines.

## Impact of bi-functional AREs on RNA stability

AREs are well-recognized regulatory motifs controlling RNA stability. Accordingly, we examined if the regulatory effect of AREs could be captured by our MPRA approach. Multiple approaches have revealed AREs as exerting a destabilizing effect on RNA stability (*Barreau et al., 2005*). However, ARE motifs and ARE-binding proteins are diverse, so the impact of binding may vary considerably. Therefore, we examined the effect of AREs on RNA stability of the ref alleles according to specific sequence content. Based on the definition of AREsite2 (http://nibiru.tbi.univie.ac.at/AREsite2), we categorized AREs as either WUUUW or its longer derivatives, UUUGUUU or AWUAAA (W:A/G; *Fallmann et al., 2016*). We observed that AREs in either the 5' or 3' UTRs generally destabilized RNA (*Figure 2A*; *Figure 2—figure supplement 1*; *Supplementary file 2*). More specifically, AUUUA/AUUA-containing AREs are associated with RNA destabilization when present in either UTR type, whereas in SH-SY5Y cells, extremely U-dominant AREs ($U_{8-10}A_{1-2}$) stabilized it (*Figure 2B*), similar to the stabilizing effect of U-stretches described for zebrafish (*Rabani et al., 2017*; *Vejnar et al., 2019*). These results suggest that although mostly destabilizing, AREs can play dual roles in regulating RNA stability by recruiting binding proteins of diverse functions.

## Modeling the impact of UTR-mediated regulation on RNA stability

Given our discovery that the effect of AREs is heavily dependent on sequence content, we decided to further explore the effects of other sequence elements, that is beyond known regulatory motifs, in more detail. Since most reported RBP motifs are 6-mers, we initiated a search for novel motifs by analyzing the presence of all 7-mers in our massively parallel reporter assay (MPRA) library, correlating their occurrence with mRNA half-life. For those with significant stabilizing or destabilizing effects, we clustered similar ones into motifs (*Figure 3—figure supplement 1*). The motifs suggest a G-rich stabilizing profile and an A-rich destabilizing profile, with the latter being more pronounced for the 3' UTR. Next, to gain a comprehensive understanding of the contextual effect of each sequence element, we took advantage of LASSO regression, which minimizes coefficients of explanatory factors to select the most influential factors. We considered as many factors as possible to explain the half-life of our ref UTR libraries, including primary sequences, RBP binding sites (ATtRACT database *Giudice et al., 2016*), miRNA seed sites, secondary structures, and folding energy. Furthermore, to avoid collinearity confounding our model, for example the effects of very similar factors (such as 'AA' and 'AAA' sequences), we clustered the factors according to their properties, and then only one representative factor from within a cluster (i.e. the one with the highest correlation to half-life within a cluster) was subjected to LASSO regression (*Figure 3A*, *Figure 3—figure supplement 2*, *Supplementary file 3* and Methods). LASSO regression renders as zero the coefficients of factors with minimal explanatory power (see Methods for details). Overall, we started with 1231 (5' UTR) or 1475 (3' UTR) factors, but

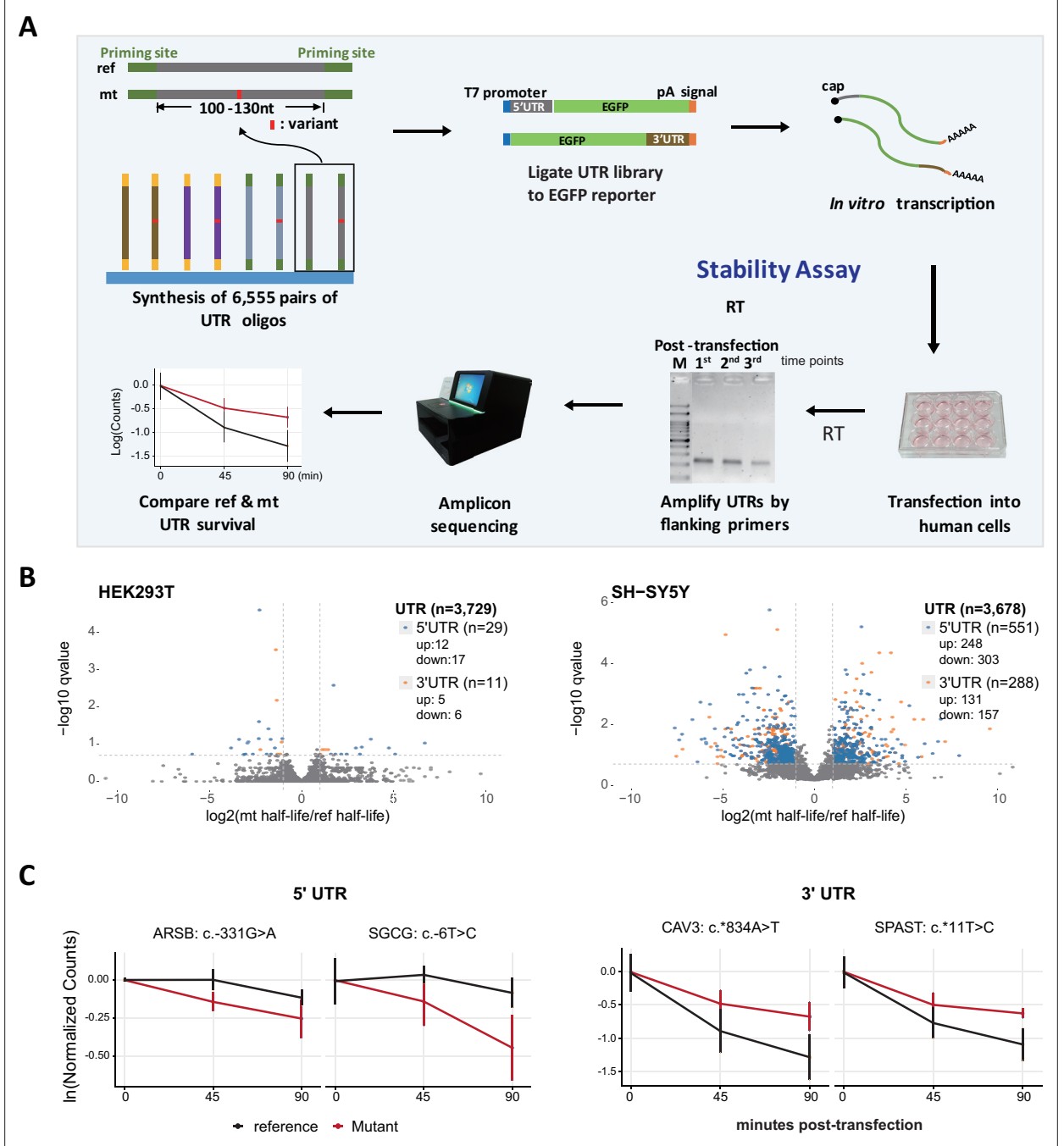

**Figure 1.** Massively parallel reporter assay (MPRA) to determine the effects of UTR variants on RNA stability. (**A**) MPRA workflow. In brief, 6555 reference (ref) and mutant (mt) UTR pairs were synthesized in bulk, ligated with promoters and reporter sequences, in vitro-transcribed into capped and tailed RNAs, transfected into human cell lines, and then the remaining RNAs were collected over a time-course. The collected RNAs were reverse-transcribed, amplified and sequenced to resolve the genotype of each UTR. The unique sequences were used to calculate RNA half-life. Mutational effects were inferred from those pairs significantly differing in half-life (see Methods). (**B**) Volcano plot of MPRA data from three repeated experiments. The colored dots indicate significant stability-altering variants. (**C**) Examples of significant stability-altering UTR mutations in both UTR types. Data are presented as mean ± SD (n = 3 experimental replicates).

The online version of this article includes the following source data and figure supplement(s) for figure 1:

**Source data 1.** UTR stability through a time course (labeled image).

**Source data 2.** UTR stability through a time course (original image).

**Figure supplement 1.** Correlation of experimental results.

*Figure 1 continued on next page*

*Figure 1 continued*

**Figure supplement 1—source data 1.** Polyadenylation of in vitro transcribed RNA (original image).

**Figure supplement 1—source data 2.** Polyadenylation of in vitro transcribed RNA (labeled image).

only 5–19 factors were selected ultimately for each trained model (*Figure 3B–E*; *Supplementary file 4*). The selected explanatory factors all represent small kmer motifs (k=2–3) and RBP binding motifs. Unexpectedly, we identified some unique regulatory factors in each cell line, indicating that RNA decay pathways are typically shared but can be strongly influenced by the cellular environment. Overall, motifs that are at least two nucleotides long proved critical for RNA stability, supporting the sequence specificity of the decay process.

In both of the cell lines we tested, GU-rich sequences in 5' UTRs stabilized RNAs (*Figure 3B and D*). In contrast, CA- and UG-repeat sequences—potential binding sites for Insulin Like Growth Factor 2 mRNA Binding Protein 3 (IGF2BP3) and CUGBP Elav-Like Family Member 1 (CELF1)—in 3' UTRs proved the most destabilizing factors in HEK293T cells (*Figure 3C*). Moreover, we noticed that for both UTRs, UA dinucleotides and other UA-rich sequences—such as WWWWWW (W=A/U; a potential binding motif of Peptidylprolyl Isomerase E (PPIE)), AUUUA (a potential binding motif of ELAV Like RNA Binding Protein 1 (ELAVL1)), and UUUAUA (a potential binding motif of hnRNPA1)—are strongly destabilizing (*Figure 3B-E* and *Figure 4A,C*). UA dinucleotides and WWWWWW belong to the same cluster, but they were respectively selected for LASSO regression in the two cell lines because they displayed the highest explanatory power (largest coefficient by univariate regression) for RNA half-life in each cell line (*Figures 3A, 4B and D*). Most prominently, UA dinucleotides in both UTRs overwhelmed other factors in robustly destabilizing RNAs in SH-SY5Y cells (*Figure 3D and E*). Therefore, UA dinucleotides seem to be a universal destabilizing motif, so we investigated how UA dinucleotides regulate RNA stability in further detail.

## UA dinucleotides and UA-rich motifs are the most common and effective RNA destabilizing factor

UA dinucleotides proved to be the strongest stability determinant for both UTR types in SH-SY5Y cells. UA dinucleotides alone present a negative correlation with RNA stability, with a Pearson's correlation coefficient of –0.287 for 5' UTRs and –0.377 for 3' UTRs (*Figure 4A and C*). UA-rich motifs (in the same cluster as UA dinucleotides) behave similarly to UA dinucleotides in regulating RNA stability, whereas GC-rich motifs have the opposite effect (*Figure 4B and D*; *Figure 4—figure supplement 1A and B*). Within the same cluster, UA dinucleotides and the WWWWWW motif were the strongest RNA stability regulators in each cell line (*Supplementary file 3*). Given the strong destabilizing effect of factors in the UA-associated cluster for both UTR types and both cell lines, we further analyzed their commonalities. An UpSet analysis revealed that all features contributing to RNA stability across four experimental groups (HEK293T 5' UTRs, HEK293T 3' UTRs, SH-SY5Y 5' UTRs, SH-SY5Y 3' UTRs) occur in the UA dinucleotide/WWWWWW cluster (*Figure 4—figure supplement 1C*), indicating a universal destabilizing effect of UA-rich sequences. Next, to examine if there is a region-specific effect of UA and closely-related AU dinucleotides, we used a sliding window to establish the localization-associated relationship between the UA/AU dinucleotide ratio and RNA half-life. Correlation coefficients between UA/AU dinucleotide ratios and UTR stability were calculated for each window, and we assumed that regions displaying a strong correlation between UA/AU dinucleotide ratios and stability rank hosted UA/AU dinucleotides that control RNA stability (*Figure 4—figure supplement 1D*). We found that UA/AU dinucleotides in the UTRs of SH-SY5Y cells were generally strongly correlated with RNA stability, but only weakly associated with RNA stability in HEK293T cells (apart from a relatively strong correlation at the ends of 3' UTRs, implying a protective role against exonuclease digestion) (*Figure 4—figure supplement 1E*). Together, these results support that UA dinucleotides are a common and prominent RNA destabilizing motif.

Next, we examined the effect of mutating the most effective destabilizing UA dinucleotide (resulting in dinucleotide gain or loss) in terms of altering RNA stability. We found a clear propensity for 3' UTRs with stability loss to accumulate gain of UA dinucleotide mutations, compared to stabilizing or non-significant mutations (*Figure 4E*). To further validate the impact of UA dinucleotides, we curated a subset of oligo pairs from a 5' UTR random library (*Jia et al., 2020*) with the sole difference

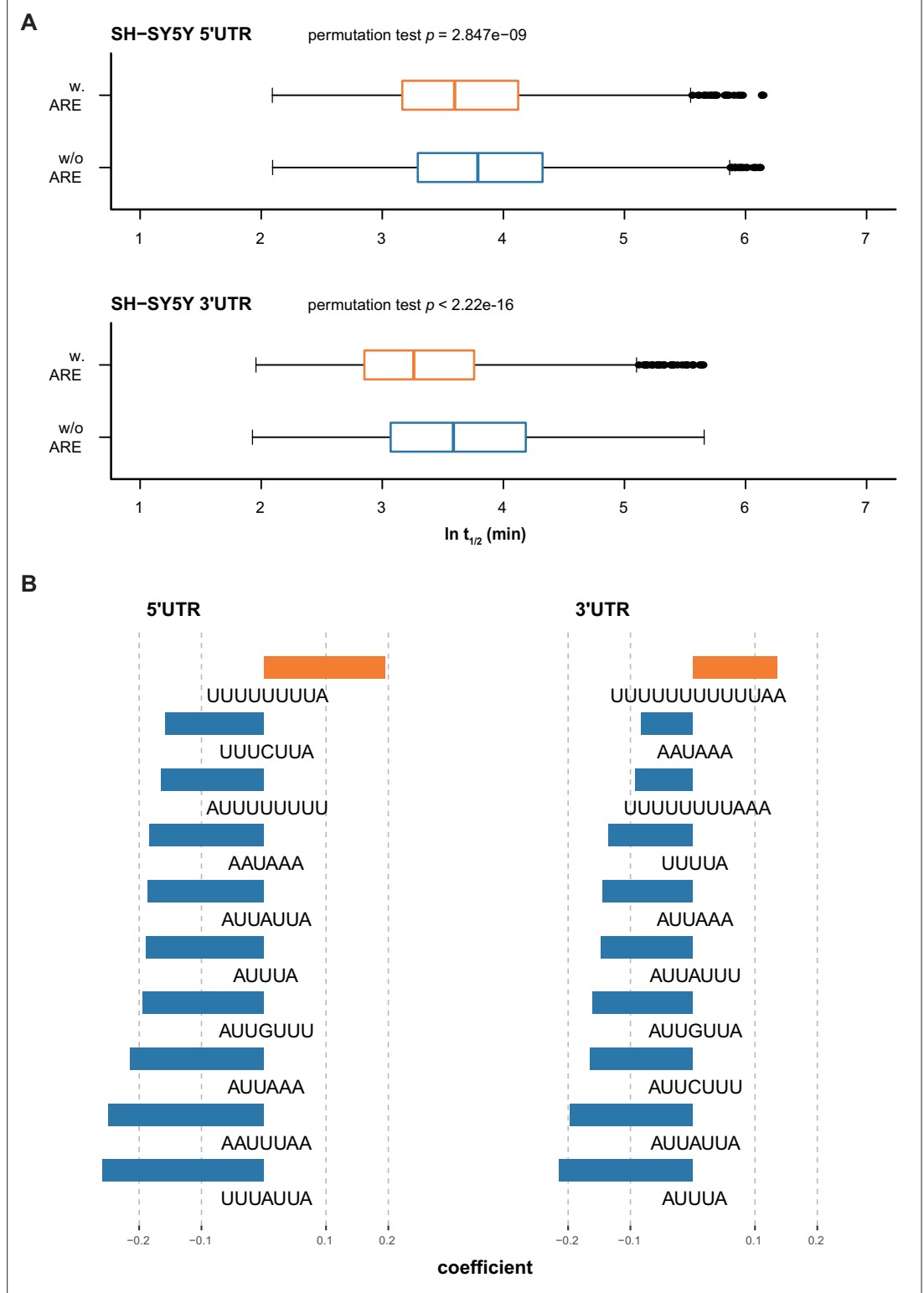

**Figure 2.** AREs generally destabilize RNAs (except extremely U-rich AREs). (**A**) AREs of both UTR types destabilize RNA. (**B**) The ten most influential AREs in terms of RNA stability in SH-SY5Y cells. Coefficients are determined by regression analysis, representing the effect size of each motif.

The online version of this article includes the following figure supplement(s) for figure 2:

**Figure supplement 1.** Various destabilizing effects of AREs in HEK293T, related to *Figure 2*.

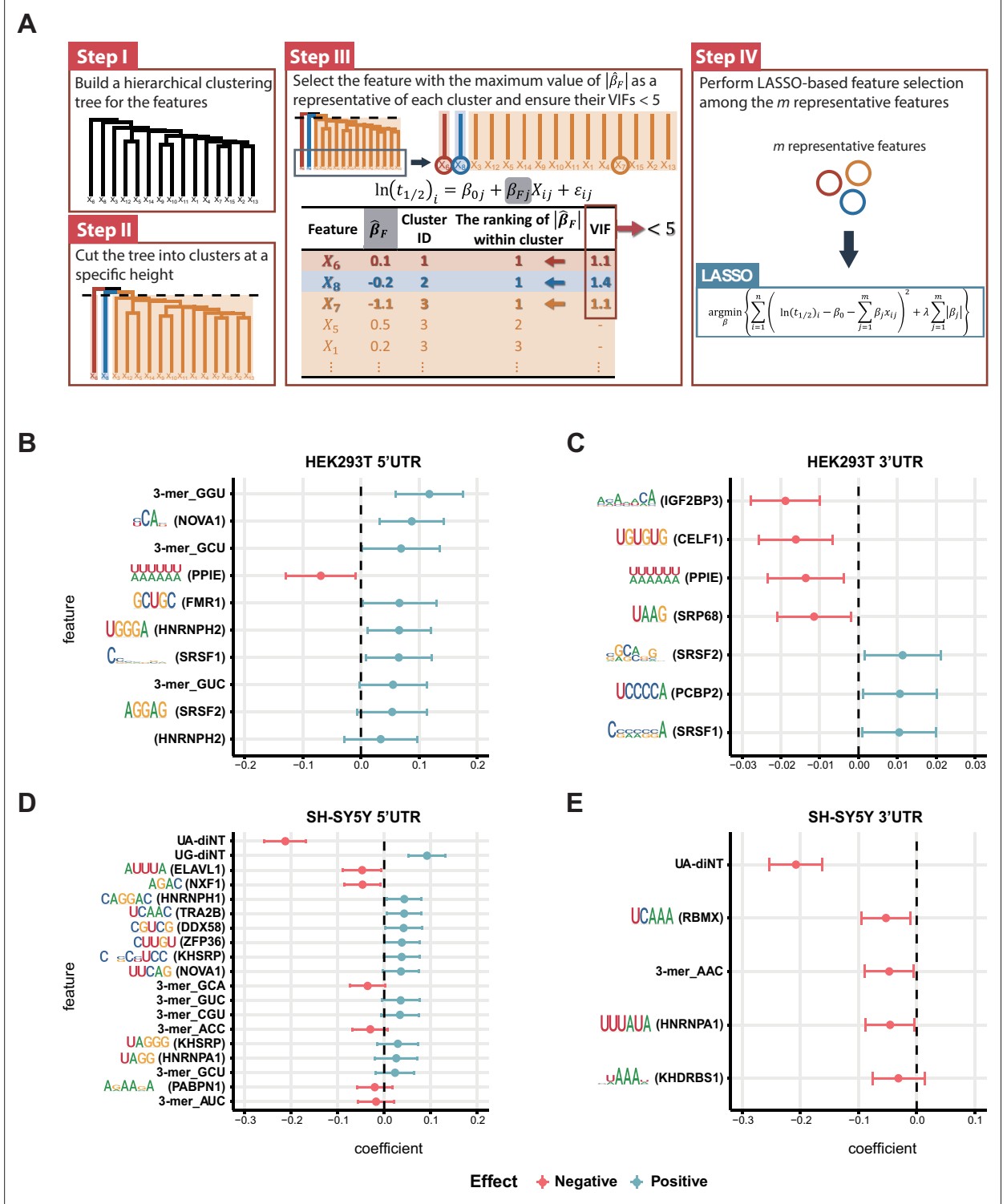

**Figure 3.** Inferential statistical analysis of RNA stability determinants. (**A**) Workflow of variable selection to build models of influencers of RNA stability. (**B–E**) Influential regulators for the 5' UTR library from HEK293T cells (**B**), the 3' UTR library from HEK293T cells (**C**), the 5' UTR library from SH-SY5Y cells (**D**), and the 3' UTR library from SH-SY5Y cells (**E**). The error bars represent 95% confidence intervals of the coefficients. Note that the factors presented on the figure are representative of their respective clusters (see Methods and *Figure 3—figure supplement 2*).

The online version of this article includes the following figure supplement(s) for figure 3:

**Figure supplement 1.** Stabilizing or destabilizing motifs in UTRs.

**Figure supplement 2.** Workflow of variable selection to build models of stability influence.

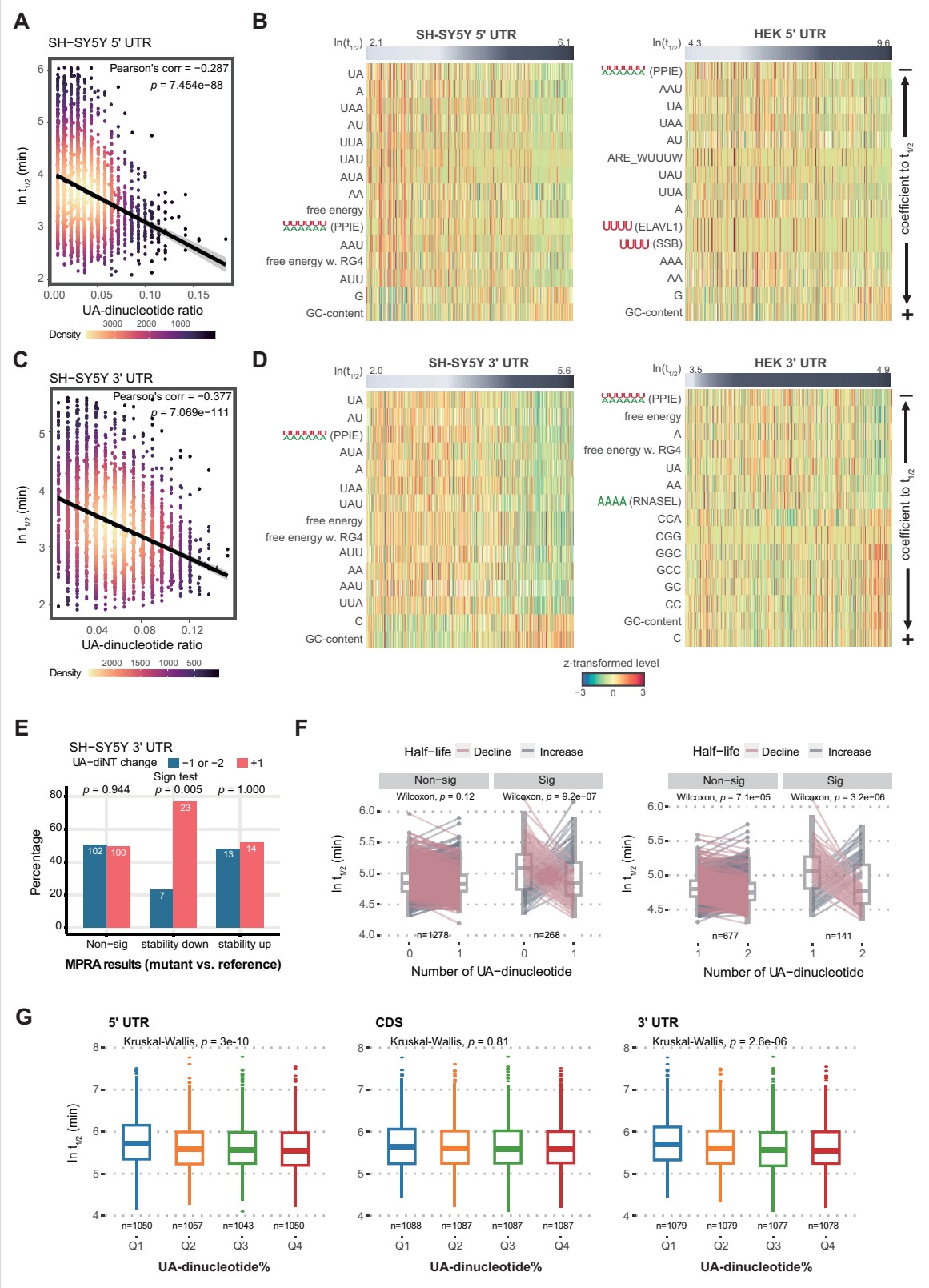

**Figure 4.** The UTR UA dinucleotides and UA-rich motifs are the most common and influential RNA destabilizing factor. (**A**) Correlation of the 5′ UTR UA dinucleotide ratio and half-life. (**B**) Top 15 influential factors in the UA cluster of 5′ UTR. UTRs are arranged by half-life, and factors by their coefficient to half-life. Note that there are destabilizing factors (such as UA and AU dinucleotides) as well as stabilizing factors (such as GC content and G monomers) in this cluster. UA dinucleotide and WWWWWWW (PPIE) (where W represents A/U) are representative of the cluster for modeling UTR stability in SH-SY5Y

*Figure 4 continued on next page*

*Figure 4 continued*

and HEK293T cells, respectively. (**C**) Correlation of the 3' UTR UA dinucleotide ratio and half-life. (**D**) Top 15 influential factors in the UA cluster of 3' UTR. (**E**) Mutational gain of a UA dinucleotide by 3' UTRs significantly reduces RNA stability (lower panel). (**F**) Gain of UA dinucleotides in a random 5' UTR library led to RNA destabilization. We categorized pairs with a≥1.5 fold change as 'significant' (Sig) and those with less than this threshold as 'non-significant' (Non-sig). (**G**) High UA-nucleotide ratios of both UTRs reduce endogenous RNA stability in HEK293 cells. Q1-Q4 denote quantile groups categorized based on the UA-dinucleotide ratio.

The online version of this article includes the following figure supplement(s) for figure 4:

**Figure supplement 1.** UA cluster is the most potent destabilizing factor in each cell line.

being the presence of one additional UA dinucleotide, resulting in a discrepancy of one UA dinucleotide between them. This selection allowed us to investigate whether these variations influenced RNA half-lives. We defined significant pairs as those with half-life differences greater than or equal to a 1.5-fold change. Notably, the acquisition of an additional UA dinucleotide resulted in the destabilization of RNAs (*Figure 4F*, Left: 0–1 UA dinucleotide; Right: 1–2 UA dinucleotides). Moreover, to explore the influence of the UA-destabilizing effect on endogenous mRNA stability, we assessed the UA-dinucleotide ratio in relation to RNA half-life of HEK293 and human erythroleukemia K562 cells. While exon junction density and transcript length have been suggested as major determinants of RNA half-life in vivo (*Agarwal and Kelley, 2022*; *Blumberg et al., 2021*), our findings indicate that elevated UA-dinucleotide ratios in both UTRs significantly promote RNA decay (*Figure 4G* and *Figure 4—figure supplement 1G*). This effect is specific, as such ratios in the coding region are inconsequential. Thus, we have identified UA dinucleotides as being the strongest RNA destabilizing feature in UTRs, with their mutational addition proving the most prevalent cause of reduced RNA stability.

## Intrinsic features of UTRs determine RNA stability

Apart from their shared regulatory mechanisms, our MPRA revealed distinct UTR behaviors. Despite undergoing exactly the same procedure, library RNAs in SH-SY5Y cells degraded much faster than those in HEK293T cells (*Figure 1—figure supplement 1D*). Distinct RNA stability control in neuronal cells by GC content of transcripts has been reported (*Guvenek et al., 2022*). Additionally, cell-specific contexts, such as expression of RBPs and miRNAs, may have intensified the discrepancy in stability control between the two cell lines. Moreover, for mutant UTRs that significantly destabilized or stabilized RNAs, we found that their ref counterparts are with significantly longer or shorter RNA half-life, respectively (*Figure 5—figure supplement 1A*). This intrinsic difference in effects on RNA stability for ref UTRs was observed for both cell lines and it was amplified in their mutant counterparts (*Figure 5—figure supplement 1B and C*). A motif analysis revealed that ref UTRs whose mutant counterparts significantly altered RNA stability tended to harbor more NOVA1 (NOVA Alternative Splicing Regulator 1) and PPIE binding sites, but fewer FMR1 (fragile X mental retardation 1) binding sites (*Figure 5—figure supplement 1D*). These observations indicate that intrinsic properties of the UTRs where mutations occur greatly influence the mutational effect.

## GC content RBP- and ribosome-binding hinders the destabilizing effect of UA dinucleotides

Many previous studies have reported GC content to be a major RNA stability determinant (*Courel et al., 2019*; *Litterman et al., 2019*; *Zhao et al., 2014*). A univariate regression analysis on our UTR libraries revealed that both GC content and UA dinucleotides are strongly associated with RNA half-life, with the 3' UTR library from SH-SY5Y cells displaying the strongest association. However, overall, UA dinucleotides are more strongly correlated with RNA half-life than GC content (*Figure 4B and D*; *Figure 5—figure supplement 2A*). We reasoned that GC content and UA dinucleotides may represent confounding factors in our models. Therefore, to further dissect their respective contributions to RNA stability, we examined their relationship by regressing both factors as well as their interaction term against RNA half-life ($t_{1/2}$ ~ UA-diNT% + GC% + UA-diNT% × GC%). For both 5' UTRs and 3' UTRs, we observed that UA dinucleotides exhibited a stronger association (i.e. smaller p values) with RNA half-life than GC content. Indeed, the link between GC content and RNA half-life became non-significant when we also accounted for UA dinucleotides (*Figure 5—figure supplement 2A*). Moreover, the interaction term (UA-diNT% × GC%) proved

significant for three out of four experimental groups (SH-SY5Y 5' UTR: p = 0.18, 3' UTR: p = 0.00047, HEK293T 5' UTR: p = 0.0062, 3' UTR: p = 1.2e-7), supporting that GC content influences the effect of UA dinucleotides. To further explore this interaction, we stratified degrees of GC content and examined their effects on UA dinucleotides. Notably, despite a somewhat anti-correlation between UA dinucleotides and GC content, they do not simply oppose each other. Substantial counts of UA dinucleotides are observed in regions characterized by high GC content. We found that UA dinucleotides strongly destabilized RNA in the context of low GC content (bottom 50%), but their effects were neutralized somewhat under scenarios of high GC content (top 50%; *Figure 5A*). The GC protective effect also supports the observation that change of UA dinucleotides in high GC-content 5' UTR did not always translate into a change of stability (*Figure 4F*). Conversely, the protective effect of GC content was remarkably strong for high UA-dinucleotide ratios but was barely detectable for low UA-dinucleotide ratios (*Figure 5—figure supplement 2B*). Thus, the UA destabilizing effect is most pronounced under conditions of a low UA-dinucleotide ratio and low GC content (*Figure 5—figure supplement 2C*). Moreover, the stabilizing or destabilizing effects of UA deletion or addition, respectively, were only observed for low GC content, further supporting that high GC content hinders the destabilizing effect of UA dinucleotides (*Figure 5B*, *Figure 5—figure supplement 2D*).

To corroborate the interplay between the UA-dinucleotide ratio and GC content, we investigated their combined impact on RNA stability using a fixed-length 5' UTR random library (*Jia et al., 2020*). Our findings reveal a negative association between the UA-dinucleotide count within the library and RNA half-life, with this effect being attenuated under conditions of high GC content (top 50%; *Figure 5C*). As another layer of validation, we further explored their relative contributions to mRNA enrichment in P-bodies (*Hubstenberger et al., 2017*). P-bodies are membraneless granules for RNA storage and turnover (*Beadle et al., 2023*). We uncovered a positive correlation between UA dinucleotides and RNA P-body localization, especially for those occurring in 3' UTRs (*Figure 5D*). For both UTR types, we observed a greater UA dinucleotide-stabilizing effect in P-bodies when GC content is low (*Figure 5D*), consistent with our MPRA dataset.

We hypothesized that the protective effect of GC content arises from extensive intramolecular interactions that shield UA dinucleotides from being recognized by nucleases, though other physical hindrances may also diminish the destabilizing effect of UA dinucleotides. Therefore, we examined the influence of RBP binding on the destabilizing effect of UA dinucleotides. In *Figure 5F and G*, we calculated the number of RBP species binding to UA dinucleotides using both experimental data (enhanced crosslinking immunoprecipitation or eCLIP) and predicted RBP-binding motifs (ATtRACT). We then organized our MPRA-derived results based on whether they exhibited a high (top 50%) or low (bottom 50%) number of RBP binding partners per UA dinucleotide. In doing so, we revealed that the RNA group with multiple binding partners indeed displayed longer half-life compared to the group with few binding partners. This result proved consistent for both UTR types and based on experimental (*Figure 5F*) or predicted (*Figure 5G*) RBP binding data. To further validate the RBP protective effect against UA-destabilization, we selected four 3' UTRs (APC, WDR35, SH3TC2, and MTR) with a UA-dinucleotide ratio greater than 90% quantile and assessed their RNA stability by transcription inhibition with actinomycin D (*Figure 5H, I*). Half-life obtained by the actinomycin D treatment were highly correlated with the MPRA result ($\rho$ =0.8) in accordance with numbers of RBP binding sites per UA dinucleotide. Together, the results argue a protective role of RBP binding on UA dinucleotides.

In acknowledgment of the dual potential of RBPs to either promote or inhibit RNA degradation, we conducted a detailed analysis of the impact of each RBP's binding on RNA stability. By correlating the binding of UA dinucleotide-binding proteins with RNA half-life, we identified UA-binding RBPs that play roles in either safeguarding or promoting RNA degradation (*Figure 5J*; *Figure 5—figure supplement 2E*; *Supplementary file 5*). Notably, the protective UA-containing motifs were found to be U-rich, reinforcing the observation that U-stretch sequences may enhance RNA stability. Thus, we have identified interplay between GC content, RBP-binding and the destabilizing effect on RNAs of UA dinucleotides. The fact that these regulatory factors may control RNA stability via synergistic or antagonistic mechanisms emphasizes the need to consider all contributory factors simultaneously, as achieved by our modeling approach, to gain a complete overview of the regulatory network (*Figure 3*).

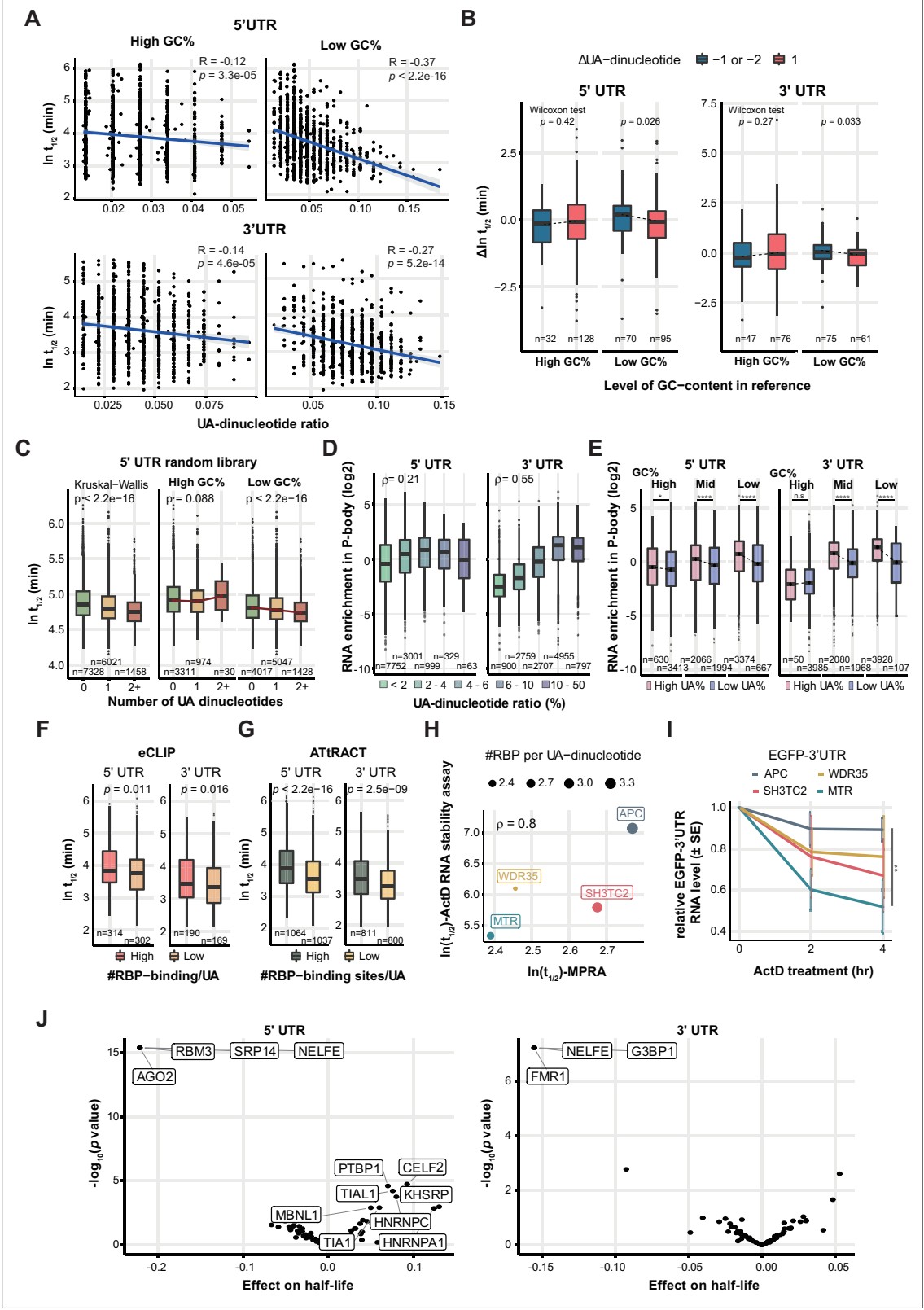

**Figure 5.** GC content, RBP and ribosome binding shields RNA from the destabilizing effect of UA dinucleotides. (**A**) MPRA data of SH-SY5Y cells stratified according to the GC content (GC%) of UTRs. The data was divided into high and low groups according to the median of GC%. In both UTRs, the destabilizing effect of UA dinucleotides is more evident in the context of low GC content (right panels). p values were determined by linear regression. (**B**) High GC content hinders the effect of altered UA dinucleotides in mutant UTRs. The destabilizing effect of UA-addition (blue) and the

*Figure 5 continued on next page*

*Figure 5 continued*

stabilizing effect of UA-deletion (crimson) are only observed under the condition of low GC content. p values were determined by a two-sided Wilcoxon rank sum test. (**C**) Destabilizing effect of UA dinucleotide is observed with 5' UTR random library. High GC content hinders the UA-destabilizing effect. (**D**) UA dinucleotides are enriched in P-body-resident mRNAs. $\rho$ represents Spearman's correlation coefficient. (**E**) High GC content inhibits enrichment of UA dinucleotide-hosting mRNAs in P-bodies. For medium or low GC%, a high UA-dinucleotide ratio promoted the P-body localization of mRNAs, but this was not the case for high GC%. This effect was more prominent for 3' UTRs. p values were determined by a two-sided Wilcoxon rank sum test. (**F**) UTRs with more eCLIP RBP binding signals per UA dinucleotide are more stable. The high and low groups was stratified based on the median of number of RBPs per UA. (**G**) UTRs harboring more predicted RBP-binding sites per UA dinucleotide are more stable, as determined by MPRA. p values were determined by two-sided Wilcoxon rank sum test (**F and G**). (**H**) Comparison of RNA half-life of high-UA UTRs determined by MPRA and transcription inhibition with actinomycin D (ActD) (**H**). $\rho$ represents Spearman's correlation coefficient. (**I**) RNA stability assay with Actinomycin D treatment. Error bars are standard errors computed from three experimental replicates. **: FDR-adjusted p-value = 0.002. (**J**) Association of UA dinucleotide-binding protein motifs with RNA half-life in SH-SY5Y cells. Note that UA-binding RBPs can have both positive and negative effects on RNA stability.

The online version of this article includes the following figure supplement(s) for figure 5:

**Figure supplement 1.** Distinct intrinsic properties associated with stability alteration.

**Figure supplement 2.** Interplay of GC content and UA dinucleotide on stability regulation, related to *Figure 5*.

## UA-dinucleotide ratio of UTRs reflects functional enrichment

Since we identified the UA dinucleotides of UTRs as a major stability determinant, we argued that if UA dinucleotides do indeed represent a functional motif regulating RNA stability, then this property could be a proxy of expression dynamics for classifying genes into functional groups. Therefore, we calculated the average UA-dinucleotide ratio of each gene and compared this distribution within a given GO term against the entire genome background. We found that the 5' UTRs of genes responsible for regulating appetite, apoptosis and synaptic signaling display the highest UA-dinucleotide ratios and those involved in glutathione metabolism exhibit the lowest (*Figure 6A*; *Supplementary file 6*). In terms of 3' UTRs, genes linked to integrin activation, the interleukin-mediated pathway and B-cell differentiation have the highest UA-dinucleotide ratios (*Figure 6B*). To investigate how UA dinucleotide localization contributes to biological functions, we utilized a sliding window approach to identify UTR regions with UA dinucleotides above or below genomic background levels (*Figure 6C and D*). UA dinucleotide in each 10-nt window with 1-nt step was calculated and normalized to the UTR length (Methods). For 5' UTRs, we found that synaptic signaling and striated muscle contraction pathways represent the functional groups with genes having most UA dinucleotide-enriched windows, whereas genes responsible for regulating cell proliferation had most UA dinucleotides-depleted windows (*Figure 6C*). For 3' UTRs, RNA translation and immune-related functions proved to be the GO terms with most UA dinucleotide-enriched windows (*Figure 6D*). Since 5' and 3' UTRs may synergistically regulate gene expression, we examined genes for which the UA-dinucleotide ratio in both UTRs significantly differed from the genomic background, which revealed that the UA-dinucleotide ratio in both UTRs was consistently either above or below background values (*Figure 6E and F*). Plotting these positive or negative effects two-dimensionally, we observed that the gene groups mostly lie in the first and third quadrants, which we interpret as indicative of a synergistic stabilizing or destabilizing effect of both UTRs. Notably, the gene groups in the first quadrant, reflecting UA-dinucleotide ratios in both UTRs being above background levels, are all related to the immune response. It is also noteworthy that the high UA-dinucleotide ratio of genes regulating appetite, synaptic signaling and the immune response reflects the transient expression nature of these gene groups, further supporting that UA dinucleotides exert a destabilizing effect on RNA. Together, this genome-wide analysis indicates that the UA-dinucleotide ratio of UTRs reflects global regulation of gene expression dynamics.

## UTR variants associated with disease

The results from our MPRA stability assay and the genome-wide functional classification support the hypothesis that UTR-mediated RNA stability and gene expression may be disrupted by SNPs within the UTRs. To expand our findings from controlled MPRA experiments to human physiological conditions, we explored the effect of genetic variations in UTRs by surveying human disease databases and biobanks.

Since dysregulated RNA stability is known to contribute to cancer progression (*Perron et al., 2022*), we first investigated UTR mutations in samples taken from cancer patients (Harmonized Cancer Datasets: https://portal.gdc.cancer.gov/). We identified several SNPs in UTRs that correlated with

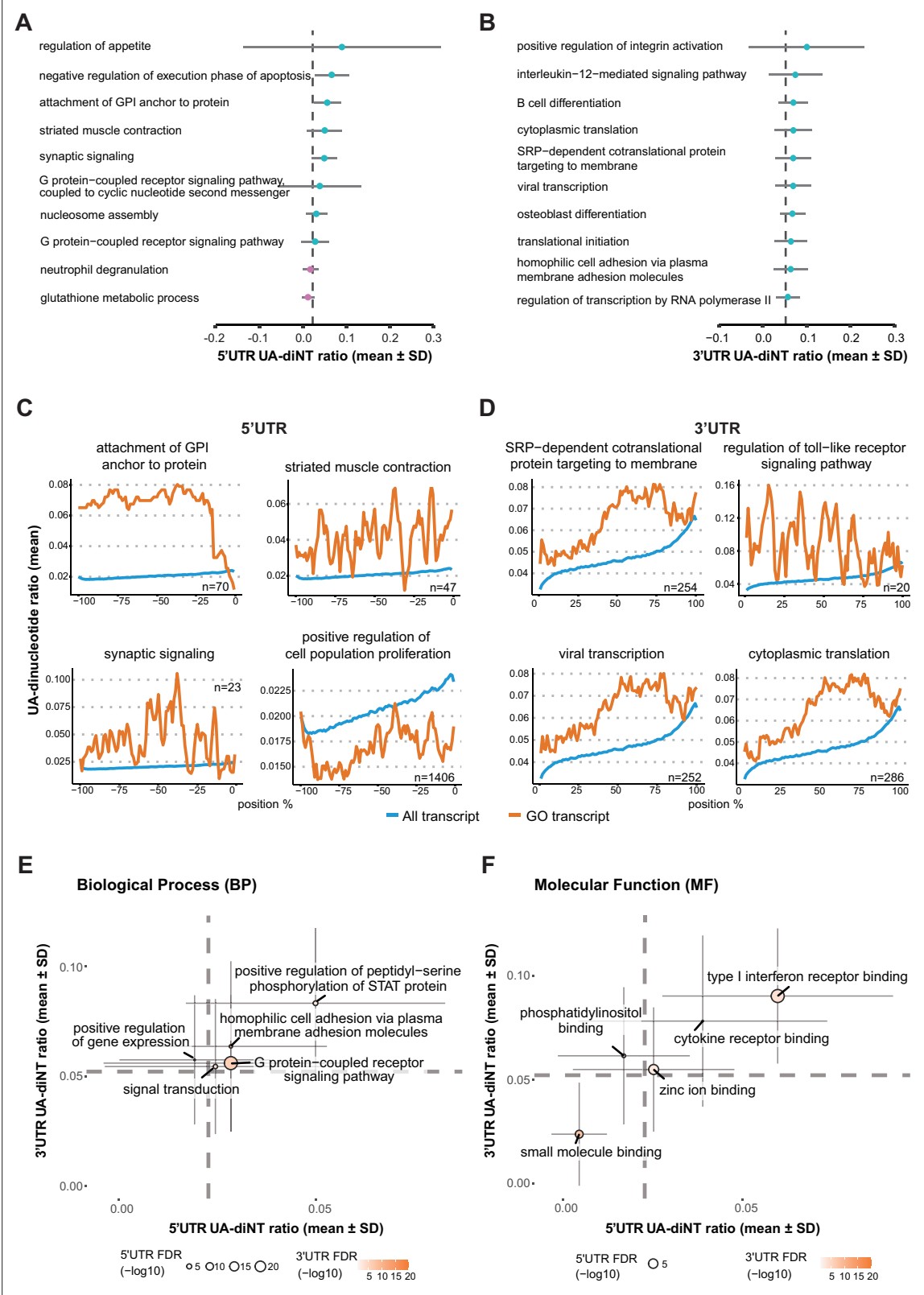

**Figure 6.** UA dinucleotides delineate functional gene groups. (**A**) The top ten biological processes for which the 5′ UTR UA-dinucleotide ratio most significantly deviated from the genomic background (dashed line). (**B**) The top ten biological processes for which the 3′ UTR UA-dinucleotide ratio most significantly deviated from the genomic background. (**C**) Functional gene groups for which the 5′ UTR UA-dinucleotide ratio was significantly above or below the genomic background in more than ten sliding windows. (**D**) Functional gene groups for which the 3′ UTR UA-dinucleotide ratio was

*Figure 6 continued on next page*

*Figure 6 continued*

significantly above or below the genomic background in more than ten sliding windows. (**E**) Biological processes for RNAs in which the UA-dinucleotide ratios of both 5' and 3' UTRs are significantly different from the genomic background (dashed lines). (**F**) Molecular functions for RNAs in which the UA-dinucleotide ratios of both 5' and 3' UTRs are significantly different from the genomic background (dashed lines). The thin solid lines represent the standard deviation of the UA-dinucleotide ratio within the gene group.

aberrant RNA expression and/or protein expression (*Supplementary file 7*). Interestingly, two 3' UTR mutations that resulted in aberrant expression of both RNA and proteins (*DDP4* [Dipeptidyl Peptidase 4] and CASP7 [Caspase 7], respectively) were A/T deletions in TA-rich sequences with increased RNA and protein expression levels (*Figure 7A and B*), in agreement with our findings that UA-rich sequences are the most influential stability determinants (*Figure 3*).

Finally, to establish a correlation between UTR variants and health outcomes, we examined if stability-altering UTR variations identified in our MPRA experiments are associated with abnormal physiological/pathological presentations among the Taiwanese community (Taiwan Biobank (TWB) data). We employed a quantile-quantile (Q-Q) plot to compare the p-value distribution of significant stability-altering UTR SNPs associated with skewed biochemical indices or self-reported diseases against theoretical p values (*Figure 7C*). We observed that p values were skewed towards the stability-altering variants, indicating that TWB subjects harboring stability-altering UTR variants are more likely to display abnormal biochemical phenotypes. The most significant association was detected for the 3' UTR of APOC3 (apolipoprotein C3 c.*40G>C) and blood triglyceride levels ($p = 3.5 \times 10^{-74}$ as determined by linear regression) (*Figure 7C and D*), total cholesterol ($p = 7.6 \times 10^{-12}$) and self-reported hyperlipidemia (p = 0.00038; *Supplementary file 8*). APOC3 is involved in metabolizing triglyceride-rich lipoproteins, and its mutation has been associated with low plasma triglyceride levels (*Borén et al., 2020*; *Goyal et al., 2021*). Our findings provide compelling evidence that regulation of RNA turnover by UTRs controls metabolic equilibrium, which can be perturbed by SNPs within the UTRs. Overall, we have demonstrated that pathogenic UTR variants are enriched in critical regulatory regions and may elicit disease.

## Discussion

### Small kmers in UTRs determine RNA stability

The function of UTRs in regulating RNA stability has long been recognized. However, very few reports have systematically addressed the functional impact of genetic variations in UTRs (*Griesemer et al., 2021*; *Sample et al., 2019*). Therefore, UTR variants are typically classified as being benign or of unknown functions, without experimental support. To methodically dissect the impact of such UTR variants, we have established an MPRA to test the effect of disease-related UTR variants on RNA stability. Unlike previous assays that measure the RNA:DNA ratio or protein output to infer RNA stability, our assay directly measures RNA survival and thereby circumvents confounding effects on transcription and/or protein production. However, while our approach effectively assesses the stability of synthesized RNA in human cells, it may not fully capture the decay dynamics of nuclear-synthesized RNA, which can be influenced by endogenous modifications and *trans*-acting RNA binding factors.

From among almost 1500 potential stability regulators, we applied LASSO regression to select 5–19 independent factors that best explained RNA half-life. The major destabilizing factors proved to be UA dinucleotides, as well as WWWWWW (W:A/U) and AUUUA (ELAV1 binding site) motifs, all highlighting the importance of UA-rich sequences in UTRs for RNA stability (*Figure 3*). Among these destabilizing factors, UA dinucleotides were best correlated with RNA half-life (*Figure 4A–D*). The UA-dinucleotide ratio rather than GC content or folding energy explained our RNA stability data better, implying that specific sequence recognition by *trans* factors and not simply regulation according to structuredness is the underlying control mechanism. Similarly, MPRA on 3' UTR variants identified mono- or di-nucleotide composition as primary factors controlling RNA expression (*Griesemer et al., 2021*). This di-nucleotide specificity may partially be attributed to the sequence preference of a human ribonuclease superfamily in which eight catalytically-active RNases (numbered 1–8) all share homology with bovine pancreatic ribonuclease A. RNase A cleaves 3'–5' phosphodiester bonds with a specificity for pyrimidines (U/C) at the main anchoring site and purines (A/G) at the secondary site (*Sorrentino, 2010*). Kinetic analysis has revealed a >100-fold preference for UpA than

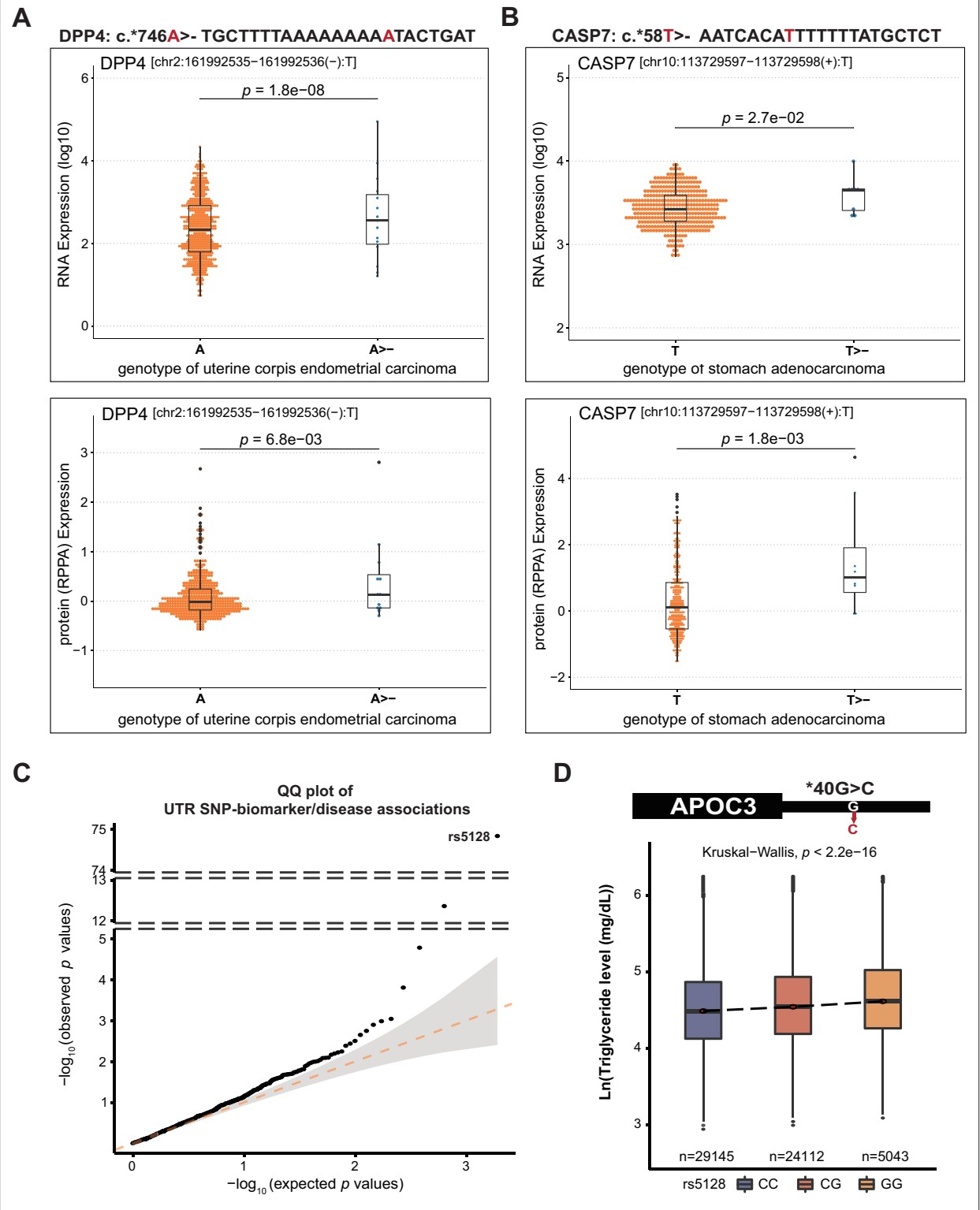

**Figure 7.** UTR variants associated with disease. (**A–B**) 3′ UTR mutations that increase RNA and protein expression in carcinoma samples. Protein level was determined by reverse phase protein arrays (RPPA). (**C**) QQ plot of the p value distribution of stability-altering UTR variants in association with health biomarkers or self-reported diseases against a theoretical distribution. (**D**) The G allele of the most significant UTR variant (rs5128) identified in (**C**) is associated with plasma triglyceride levels in the Taiwanese population (TWB dataset).

UpG substrates for the human RNase A family (RNase 1–7, RNase 8 was not studied in this report) (*Prats-Ejarque et al., 2019*), consistent with our finding that UA dinucleotides represent the most destabilizing sequence motifs. While most of the studies on RNA degradation mechanism concerned deadenylation and decapping followed by exonuclease digestion, the contribution of endonucleases on overall RNA degradation might be underestimated. It was reported that transcript and UTR length is negatively correlated with RNA stability (*Neymotin et al., 2015*; *Blumberg et al., 2021*), implying that increased RNA mass could be susceptible to endonuclease attack. Thus, our findings and those of others demonstrate that specific sequence recognition, especially di-nucleotide composition, determines RNA stability.

## Interplay of GC content, RBP binding and UA dinucleotides for RNA stability

Although it was intuitive to infer a negative correlation between UA dinucleotides and GC content, the best-known stability regulator, we found that UA dinucleotides cannot simply be viewed as an inverse proxy for GC content. Instead, the UA-dinucleotide ratio more adequately explained our RNA stability data, and the protective effect of GC content on stability was almost abolished in the context of a low UA-dinucleotide ratio (*Figure 5—figure supplement 2*). Consequently, UA dinucleotides destabilize RNA more robustly under conditions of low GC content (*Figure 5A*; *Figure 5—figure supplement 2*). This reciprocal interaction was also apparent in independent RNA stability assays and P-body transcriptomic analysis that inferred RNA degradation in vivo (*Hubstenberger et al., 2017*; *Jia et al., 2020*; *Figure 5C–E*). Thus, the effect of GC content was also impacted by the sequence context, potentially explaining discrepancies among previous reports on the effects of GC content on RNA stability (*Courel et al., 2019*; *Griesemer et al., 2021*; *Litterman et al., 2019*; *Zhao et al., 2014*). In contrast, although RBPs do not always protect RNA from degradation, with their effect on RNA stability depending on the factors recruited (*Figure 5J*), the destabilizing effect of UA dinucleotides is generally hampered by RBP binding (*Figure 5F and G*). We reason that, similar to structural hindrance, recognition of UA dinucleotides can be blocked by the physical occupancy of RBPs, resulting in an overall protective effect of RBPs against UA dinucleotide-mediated degradation that may overwhelm the destabilizing effect of a subset of ARE-binding proteins.

## UTR variants linked to RNA stability and population health

We identified many crucial regulatory features in 5′ UTRs. Thus, our results evidence that 5′ UTRs are an indispensable region for controlling RNA stability. Our MPRA data revealed that more 5′ UTR variants than those in 3′ UTRs caused stability changes (*Figure 1B*). Moreover, we found that stability-altering variants of both UTR types are associated with skewed biomarker or disease accessions in the TWB. Although we did not intend to dissect translation-dependent or -independent decay pathways in our system, it demonstrated the critical contribution of both UTR types to regulating RNA stability. Significantly, our presentation of UA dinucleotide enrichment or depletion in the UTRs of functional gene groups indicates that both UTR types may jointly regulate RNA stability to achieve kinetic control of a functional pathway (*Figure 6E and F*). At the sequence level, our results indicate that mutational gain of UA dinucleotides is linked to diminished RNA half-life (*Figure 4E*). This simple rule can be adopted as a primary screening for UTR mutation-mediated pathologies and a principle for sequence design of synthetic RNA to be expressed in human cells. Further validation of the sequence effects can be undertaken in pertinent cell or tissue types. Collectively, our findings have uncovered the RNA stability regulation exerted by the sequence composition of UTRs, revealed health-related UTR variants, and provided a foundation for precise diagnosis of non-coding genetic variants.

## Methods
### Metagene analysis

Both variants on all UTR and uORF use dbSNP version 151 (*Sherry et al., 1999*) as baseline, disease variants, UTR sources are as described above. uORF definitions were downloaded from TIS-db (*Wan and Qian, 2014*), and their genome coordinates were lifted from hg19 to hg38 before analysis.

## Disease variant collection, UTR library construction

### Variant collection

Disease-related variants were collected from ClinVar (*Landrum et al., 2016*) and HGMD database (*Stenson et al., 2017*). Variants located in the coding region or labeled as benign variants were removed. The UTRs were defined by NCBI RefSeq and ENCODE V27. We then intersected selected variant positions with the UTR regions by using BEDTools (*Quinlan and Hall, 2010*) to collect UTR variants.

### Library construction

For each variant, we extracted 115 bp of sequence around variant position, and built one oligo pair, reference and mutant, which only differs only by the variant position. The variant was placed in the middle of the sequence, unless it is located near the boundary of UTR. UTR-specific primers were then added on each sequence. (5' UTR-F: 5'-CGCTAGGGATCCTCTAGTCA-3', 5' UTR-R: 5'-ACCGGTCG CCACCATGGTGA-3'; 3' UTR-F: 5'-GGACGAGCTGTACAAGTAAA-3', 3' UTR-R: 5'-GCGGCCGC GCAATAACTAGC-3'). Overall, we built an oligo library containing 12,472 sequences (6555 pairs) of both 5' and 3' UTR, and synthesized them by CustomArray Inc (U.S.).

UTR-library DNA templates were assembled by overlap extension PCR with Herculase II Fusion Enzyme (Agilent Technologies). Initially, the oligonucleotide library sequences were double-strandized and amplified by PCR. After being subjected to PCR clean-up (QIAGEN), the UTR-library amplicons were then appended with EGFP CDS (derived from EGFP-N1 vector) and T7 promoter sequence through overlapping PCR. Eventually, the assembled full-length DNA templates (T7-5' UTR-EGFP or T7-EGFP-3' UTR) were subjected to PCR clean-up again for the following in vitro transcription.

### In vitro transcription and polyadenylation

200 ng of the PCR product was subject to in vitro transcription using MEGAscript T7 Transcription Kit (Thermo Fisher Scientific). To cap the RNA product, m7G(5')ppp(5')G RNA Cap Structure Analog (New England Biolabs, 4:1 to GTP) was supplemented to the in vitro transcription reaction. After 3 hr incubation in 37 °C, DNA template was removed by 1 µl Turbo DNase at 37 °C for 15 min. The RNA product was purified by illustra microspin G-50 column (GE Healthcare Life Sciences). 10 µg of the purified RNA was then polyadenylated by 4 U poly(A) polymerase (New England Biolabs) at 37 °C for 1 hr and purified again by illustra microspin G-50 column or Direct-zol RNA Miniprep kits (Zymo Research) before transfection (*Figure 1—figure supplement 1C*).

### Cell lines

HEK293T or SH-SY5Y cells were authenticated and purchased from ATCC (American Type Culture Collection), catalog numbers CRL-11268 and CRL-2266. They were tested negative for mycoplasma contamination.

### Transfection

Seed HEK293T or SH-SY5Y cells 20 hr before transfection. Transfect 500 ng capped and polyadenylated RNA per well into a 12-well plate by Lipofectamine 3000 reagent (Thermo Fisher Scientific). Wash the cells twice before collecting the first time point (30 min for HEK293T, 20 min for SH-SY5Y), and then harvest RNA along the time course.

### Amplicon preparation for sequencing

RNA was extracted using Trizol reagent (Invitrogen) and total RNA was extracted with Direct-zol RNA Miniprep kits (Zymo Research). Afterwards, cDNA was reverse transcribed from 1 µg of RNA using library-specific primers (5' UTR: EGFP3-f_UMI_5Lib-RT; 3' UTR: CMV3-f_UMI_M13-rev, see *Supplementary file 9*) with SuperScript IV Reverse Transcriptase (Thermo Fisher Scientific). The cDNA was amplified first by primers EGFP3-f and CMV3-f 15 cycles by Phusion High-Fidelity DNA Polymerase (Thermo Fisher Scientific) with GC buffer. The resultant product was cleaned up using QIAGEN PCR Purification kit and subject to second round of PCR with primers Forward/P5_fractionsID_CMV3f and Reverse/P7_#N_EGFP3f for 5' UTR library, and Forward/P5_fractionsID_EGFP3f and Reverse/

P7_#N_CMV3f (**Supplementary file 9**) for 3' UTR library by 10 cycles. The PCR product was purified again by QIAGEN PCR Purification kit for Illumina NextSeq paired-end 150 sequencing.

## RNA-seq data processing

### QC

PCR duplication of single-end raw fastq files (150 bp) was first removed by nubeam-dedup (**Dai and Guan, 2020**), then adapter and low-quality base (sliding mean quality <20) was trimmed by trimmomatic (**Bolger et al., 2014**). Trimmed reads of length less than 80 bp were discarded.

### Alignment

FASTQ data alignment was done by HISAT2 (**Kim et al., 2015**), genome information was built from the UTR library.

Detail parameters are as below, HISAT2: hisat2 `--no-spliced-alignment --score-min L,0,–`
0.7 -x index -U trimmed_fastq.gz.

### Count

After alignment, BEDTools multicov (**Quinlan and Hall, 2010**) was used to build the count matrix. Since accuracy is critical in this project, alignment results were filtered by mapping quality (HISAT2: MAPQ = 60), and count only 'exactly one alignment' result.

## Half-life estimation

For each oligonucleotide, we estimated decay constant $\lambda$ and half-life ($t_{1/2}$) by the following equations:

$$\ln\left(\frac{C_{i(t)}}{C_{i(t=0)}}\right) = -\lambda t_i + \epsilon_i$$

$$t_{1/2} = \frac{\ln(2)}{\lambda}$$

where $C_{i(t)}$ and $C_{i(t=0)}$ are read count values of the $i^{th}$ replicate at time points $t$ and 0, and $\epsilon_i$ is error term. For a more accurate estimate of $\lambda$, we used the first time point (HEK293T: 30 min, SH-SY5Y: 20 min) as the time point of t=0, and the models were weighted by their inverse standard deviations. All the read count values were normalized by the average of $C_{i(t=0)}$ before constructing the linear models. After that, half-life $\lambda$ was estimated by linear regression function 'lm' of R. We only selected oligonucleotide that $R^2$ >0.5 and mean squared error (MSE) < 1 for further analysis.

## Statistical method to detect the mutation effect on stability

To determine variants that changed oligonucleotide stability, we used the normalized counts to build a linear model weighted with the inverse of their standard deviations as follows:

$$\ln\left(\frac{C_{i(t)}}{C_{i(t=0)}}\right) = \beta t_i + \theta t_i G_i + \epsilon_i$$

where $G_i$ represented reference (ref) or mutant (mt), and $\epsilon_i$ is error term.

We tested the significance of $\theta$ to determine the effect of the variant. We then corrected the p value by FDR.

All statistical analyses were performed by R language (version 4.0.5/4.2.0), linear models were constructed by 'lm' function and p value adjustment was conducted by 'p.adjust' function and 'qvalue' function from the R package 'qvalue'.

## in silico feature prediction

### Free Energy

Free Energy was estimated by the ViennaRNA package RNAfold command (**Lorenz et al., 2011**), which predicted minimum free energy (MFE) by default.

### GC content and kmer ratio

GC content, the ratio of G/C nucleotide of a sequence, and short kmer (k=1–3) were calculated by letterFrequency and oligonucleotideFrequency function of the R package Biostrings (https://bioconductor.org/packages/Biostrings).

### Motifs

For motif scanning such as PUM motif, RBP motif, ARE and novel motif, were done by Biostrings functions, countPWM or countPattern (https://bioconductor.org/packages/Biostrings). PUM motif was retrieved from *Rabani et al., 2017*. RBP motif PWMs were download from ATtRACT database (*Giudice et al., 2016*) and ARE definition followed by AREsite database (*Fallmann et al., 2016*), including ATTTA, WTTTW, WWTTTWW, WWWWTTTWWWW, WWWWWTTTWWWWW, WWWWWWTTTWWWWWW, TTTGTTT, GTTTG, AWTAAA.

### KOZAK/uAUG

KOZAK sequence was defined as in the previous study (*Hernández et al., 2019*), optimal: GCCRCCAUGG; strong: NNNRNNAUGG; moderate_H: NNNRNNAUG(A/C/U); moderate_Y: NNN(C/U)NNAUGG and weak: NNN(C/U)NNAUG(A/C/U). We scanned these motifs on the library sequence by R package Biostrings countPattern function.

### miRNA

miRNA binding site was predicted by TargetScan 7.0 (*Agarwal et al., 2015*; *McGeary et al., 2019*). miRNA expression information was downloaded from miRmine database (*Panwar et al., 2017*), only those miRNAs that have at least 1 RPM in HEK293T or SH-SY5Y cells were selected as features. Since the 6-mer seed sequence is known to be less effective (*Bartel, 2009*), only predicted 7mer (7mer-m8, 7mer-1a) and 8mer seed sequences were kept as features.

### RNA binding protein binding

eCLIP (enhanced crosslinking immunoprecipitation) data were downloaded from ENCODE (*Luo et al., 2020a*) and GSE117290 datasets (*Luo et al., 2020b*), and selected only eCLIP regions identified in two biological replicates. eCLIP signals with value <1 or p value >0.05 were removed. All genome coordinates were lifted to hg38 by liftover command (*Hinrichs et al., 2006*). RBP bindings were determined by intersecting eCLIP signals and variant coordinates by BEDTools intersect command (*Quinlan and Hall, 2010*).

### G-quadruplex (RG4)

G-quadruplex structures were predicted by the RNAfold command from the ViennaRNA package (*Lorenz et al., 2011*), with -g parameter.

### Conservation level

phastCons scores (pC4way-pC100way) were downloaded from UCSC genome browser (*Lee et al., 2022*) and intersected with UTR genome coordinates by the intersect function of BEDTools (*Quinlan and Hall, 2010*). An average phastCons score was calculated for each UTR.

### Novel 7-mer motifs

Primer sequences were first removed from the UTR library sequence. Then a 7-mer table was built by manual R script (*R Development Core Team, 2021*). UTR library half-life was transformed by log2() and the extreme values were filtered out. 7-mers occurring less than 20 times in the UTR library were filtered out. Regress 7-mers to log2(half-life) by glmnet::glmnet() LASSO selection (*Friedman et al., 2010*) to obtain the coefficients. Repeat the regression 2000 times with bootstrap resampling and select 7-mers that were selected >1600 times for further analysis. The non-zero distribution of coefficients of a 7-mer was then tested by permutation coin::oneway_test (*Hothorn et al., 2008*), and the resulting p values were adjusted by p.adjust() Bonferroni adjustment. 7-mers with non-zero coefficients (adjusted p-values <0.05) were further divided into positive and negative effects according

to their mean coefficients. To generate motifs, the text distance in each group was calculated using stringdist::stringdistmatrix() function with 'lv' distance (*van der Loo, 2014*) and clustered by hclust(). Finally, use cutree(), h=2 to subgroup the 7-mers. 7-mers of a subgroup were combined to build PWM according to the weight of their mean coefficients.

## Feature selection

A LASSO model was utilized to perform feature selection. Features with a high proportion of 0 (≥ 90%) were excluded to ensure the robustness of the selection model. In addition, to avoid multicollinearity caused by similar features that perturb feature selection, all features were clustered using single-linkage hierarchical clustering with the distance metric defined as one minus the absolute value of the Spearman correlation coefficient. We cut the tree at a specific height, and the feature that had the greatest influence on RNA stability, which was examined using a simple linear regression model, was selected to be the representative of each cluster. Then we calculated the variance inflation factor (VIF) value of the representative features. The VIFs were obtained by the following linear model and equations:

$$\hat{X}_{ij} = \hat{a}_{0(j)} + \sum_{k \neq j} \hat{a}_{k(j)} X_{ik}$$

$$R_j^2 = \frac{\sum_i (\hat{X}_{ij} - \bar{X}_{.j})^2}{\sum_i (X_{ij} - \bar{X}_{.j})^2}$$

$$\text{VIF}_j = \frac{1}{1 - R_j^2}$$

where $\hat{X}_{ij}$ and $X_{ik}$ are the estimated value of the $j^{th}$ feature and the value of the $k^{th}$ feature of the $i^{th}$ UTR (note that the $k^{th}$ feature is a feature other than the $j^{th}$ feature), $\hat{a}_{0(j)}$ and $\hat{a}_{k(j)}$ are the intercept and the regression coefficients of the linear model that regressed the $j^{th}$ feature on the other remaining features, and $\bar{X}_{.j}$ is the mean level of the $j^{th}$ feature of all UTRs.

The height for cutting the tree was gradually increased until the VIF values of all representative features were less than 5. By identifying and excluding highly collinear variables, we aimed to minimize multicollinearity and improve the accuracy of our regression models (*Akinwande et al., 2015*). Finally, the LASSO-based feature selection was conducted on the representative features (*Figure 3—figure supplement 2*). The LASSO model was carried out using R package glmnet (*Friedman et al., 2010*) and can be written as follows:

$$\underset{\beta}{\text{argmin}} \left\{ \sum_i \left( \ln (t_{1/2})_i - \beta_0 - \sum_j \beta_j X_{ij} \right)^2 + \lambda \sum_j |\beta_j| \right\}$$

where $\beta_0$ and $\beta_j$ are the intercept and regression coefficients of the $j^{th}$ feature, and $x_{ij}$ is the level of the $j^{th}$ feature of the $i^{th}$ UTR. $\lambda$ was determined through a ten-fold cross-validation process, which yielded the minimum mean cross-validated error. We performed a 2:1 split of the data into training and testing sets to validate the robustness of the model. The performance metrics, including the correlation coefficient between observed and predicted values, mean average error (MAE), root mean squared error (RMSE), mean absolute percentage error (MAPE), and R-squared, are provided in *Supplementary file 4*.

## In vivo validation

To validate UA-diNT effect in in vivo data, we collected public SLAM-seq data set GSE126523 (K562) and GSE214396 (HEK293) to estimate the half-life of endogenous transcripts. The first step was to remove adapters and low-quality sequences (sliding window average sequencing quality <20 and read length <50) by trimmomatic. SLAM-seq fastq was then aligned to transcript sequences by HISAT-3N (*Zhang et al., 2021*) with annotation file gencode.v43.primary_assembly. After alignment, 'exactly one alignment' result was further used to calculate T to C conversion by hisat-3n-table. Nucleotide positions with less than 10 coverage or potential SNPs (T to C conversion rate larger than 0.8) were removed. Each transcript T to C conversion rate were calculated by Total TtoC / (Total TtoC +Total T).

Transcripts which has at least 2 batches in each timepoint were kept for further analysis. After all time points T to C conversion rate normalized to time 0 average T to C conversion rate, normalized data then fit

$$\ln\left(\frac{N_{i(t)}}{N_{i(t=0)}}\right) = -\lambda t_i + \epsilon_i$$

$$t_{1/2} = \frac{\ln(2)}{\lambda}$$

where $N_{i(t)}$ and $N_{i(t=0)}$ are T to C conversion rate of the i[th] transcript at time points $t$ and 0, and $\epsilon_i$ is error term. R lm() function was used to estimate $\lambda$ and calculate transcript half-life $t_{1/2}$.

UA-diNT counts were calculated by R Biostrings::countPattern and sequence length by nchar(). UA-diNT ratios in each UTR sequences were UA-diNT counts normalized to the sequence length.

## RNA stability assay with actinomycin D treatment

SH-SY5Y cells were seeded approximately $4\times10^5$ cells per well in 12-well plates before the day of the transfection. The cells were transfected with 800 ng pEGFP-3' UTR plasmids by using Viromer ONE RED (Lipocalyx GmbH). 24 hr post-transfection, the cells per 12-well were equally divided into 3 wells in 24-well plate. After 16 hr incubation, cells were treated with actinomycin D (5 μg/ml) and were harvested by TRIzol at the respective time points (0, 2, 4 hr after stopping the transcription). The RNA was purified by the Direct-zol RNA Microprep kits (Zymo Research). Equal amount of the total RNA for each sample were reverse transcribed using SuperScript IV (Thermo Fisher) with random hexamer. Real-time PCR was performed with FastSYBR Green Plus Master Mix (Applied Biosystems) and Quant-Studio 12 K Flex Real-Time PCR System (Applied Biosystems). Relative mRNA abundance in different time points were normalized to time points 0 hr ($2^{-(\Delta Ct - \Delta Ct0)}$) for each pEGFP-3' UTR.

## Construction of UTR-library DNA templates

UTR-library DNA templates were assembled by overlap extension PCR with Herculase II Fusion Enzyme (Agilent Technologies). Initially, the oligonucleotide library sequences (CustomArray, Inc USA) were double-strandized and amplified by PCR. After subjected to PCR clean-up (Qiagen), the UTR-library amplicons were then appended with EGFP CDS and CMV promoter sequence (derived from EGFP-N1 vector) sequentially through overlapping PCR. Eventually, the assembled full-length DNA templates (CMVP-5' UTR-EGFP or CMVP-EGFP-3' UTR) were subjected to PCR clean-up again for the following transfection.

## UA-ratio analysis for GO annotation

GO annotations were downloaded from QuickGO (*Binns et al., 2009*) and Ensembl BioMart (*Kinsella et al., 2011*). Full Human genome sequence and transcript annotation were from R Bioconductor package BSgenome.Hsapiens.UCSC.hg38 v.1.4.4 (UCSC version hg38, based on GRCh38.p13) and TxDb.Hsapiens.UCSC.hg38.knownGene v.3.15.0. UTR length <10 nucleotides were removed from analysis.

UA-dinucleotide ratio in each UTR sequence was calculated by (UA count ÷ (sequence length - 1)). For every GO annotation containing >20 qualified transcripts, UA-ratio of those transcripts were compared with all transcripts by Fisher-Pitman permutation test (R coin package *Hothorn et al., 2008*). Bonferroni corrected p-value <0.01 was defined as significant enrichment/depletion.

For the sliding UA-dinucleotide ratio, UA-ratio of the most 5' 10-nt window was first calculated and the window was slid by 1 nt in each movement to the 3'end for each transcript. Then each UTR was normalized to length by splitting sliding UA-ratio to 100 fragments and the mean sliding UA-dinucleotide ratio in each fragment was calculated. Finally calculate the mean ratio in each fragment across all transcripts within a GO term was calculated to represent the sliding UA-dinucleotide ratio.

## TCGA data

TCGA data were downloaded by R/Bioconductor package TCGABiolinks (*Colaprico et al., 2016*) with the following query code:

RNA-seq: GDCquery(TCGA-id, data.category = "Transcriptome Profiling", experimental.strategy = "RNA-Seq", data.type = "Gene Expression Quantification",workflow.type = "STAR - Counts").

Protein: GDCquery(project = TCGA id, data.type = "Protein expression quantification", legacy = TRUE, data.category = "Protein expression", platform = "MDA_RPPA_Core").

Only donor IDs and variants valid in both RNA/Protein datasets underwent further analyses.

## The Taiwan Biobank Data, TWB

In order to identify the association between RNA stability altering variants and common chronic diseases and biochemical indices, the genotyping and phenotypic data of 68,978 Taiwanese people were obtained from the TWB (https://www.biobank.org.tw/). The information on the diseases was self-reported and collected through questionnaires. Each participant was genotyped on the Affymetrix Axiom genome-wide TWB 2.0 array containing 752,921 SNP (single nucleotide polymorphism) probes. The study was approved by the Institutional Review Board of Academia Sinica (AS-IRB-BM-19020).

We conducted statistical analyses to assess the association of 21 variants significantly affecting RNA stability with 23 reported traits and the levels of 24 biochemical indices. Before carrying out association tests, the quality control for genotyping data was performed as described previously to ensure its reliability (*Chiang et al., 2022*). Linear regression and logistic regression were performed to examine the association between each SNP and continuous biochemical index and dichotomous trait, respectively. Multinomial logistic regression followed by a likelihood ratio test was used to determine p-values for biochemical indices containing more than two levels and was fitted by R package nnet (v 7.3–17). All SNPs were tested as co-dominant genetic models. The square root of age, gender, dwelling place, and the batch of array were included in the regression models to adjust for potential confounding effects. The first 10 principal components were also included as covariates in all regression models to control the population stratification.

## Acknowledgements

We thank the Genomics Core and the Bioinformatics Core of the Institute of Molecular Biology (IMB), Academia Sinica, for performing the amplicon sequencing and for providing computing resources. We thank all members of IMB, particularly Drs. Jun-Yi Leu and Hung-Lun Chiang, for tremendous help and support. This work was supported by Career Development Award and Multidisciplinary Health Cloud Research Program of Academia Sinica (AS-CDA-108-M03 and AS-PH-109-01-3), Career Development Award of National Health Research Institute, Taiwan (NHRI-EX112-10908BC) and Excellent Young Scholar Research Grants and Ta-You Wu Memorial Award of National Science and Technology Council, Taiwan (MOST 111–2628-B-001–003 and 108–2118 M-001-013-MY5).

## Additional information

### Funding

| Funder | Grant reference number | Author |
|---|---|---|
| Academia Sinica | AS-CDA-108-M03 | Jia-Ying Su<br>Yen-Tsung Huang |
| Academia Sinica | AS-PH-109-01-3 | Jia-Ying Su<br>Yen-Tsung Huang |
| National Health Research Institutes | NHRI-EX112-10908BC | Yun-Lin Wang<br>YoonSoon Kang<br>Chien-Ling Lin |
| National Science and Technology Council | MOST 111-2628-B-001-003 | Yu-Tung Hsieh<br>Yu-Chi Chang<br>Cheng-Han Yang<br>Chien-Ling Lin |
| National Science and Technology Council | 108-2118-M-001-013-MY5 | Yen-Tsung Huang |

The funders had no role in study design, data collection and interpretation, or the decision to submit the work for publication.

## Author contributions
Jia-Ying Su, Investigation, Visualization, Methodology, Writing – original draft, Project administration, Writing – review and editing; Yun-Lin Wang, Data curation, Formal analysis, Investigation, Visualization, Methodology, Writing – original draft, Project administration; Yu-Tung Hsieh, Validation; Yu-Chi Chang, Cheng-Han Yang, Validation, Project administration; YoonSoon Kang, Project administration; Yen-Tsung Huang, Supervision, Funding acquisition; Chien-Ling Lin, Conceptualization, Supervision, Funding acquisition, Writing – original draft, Writing – review and editing

## Author ORCIDs
Jia-Ying Su ![ORCID] https://orcid.org/0000-0001-5934-5458
Chien-Ling Lin ![ORCID] https://orcid.org/0000-0002-5730-799X

Reviewer #1 (Public review): https://doi.org/10.7554/eLife.97682.3.sa1
Reviewer #2 (Public review): https://doi.org/10.7554/eLife.97682.3.sa2
Reviewer #3 (Public review): https://doi.org/10.7554/eLife.97682.3.sa3
Author response https://doi.org/10.7554/eLife.97682.3.sa4

# Additional files

## Supplementary files
Supplementary file 1. Results of the massively parallel reporter assays.

Supplementary file 2. Correlation of AREs and RNA half-lives.

Supplementary file 3. Univariable correlation of sequence features and RNA half-lives.

Supplementary file 4. LASSO feature selection for RNA half-lives.

Supplementary file 5. Correlation of UA-binding RBPs and RNA half-lives.

Supplementary file 6. Frequency of UTR UA-dinucleotides in functional gene groups.

Supplementary file 7. RNA and protein expression level in association with genotypes in cancers.

Supplementary file 8. Correlation tests of health markers and genotypes in TWB.

Supplementary file 9. Primer list.

MDAR checklist

## Data availability
All raw and processed sequencing data generated in this study have been submitted to the NCBI Gene Expression Omnibus (GEO; https://www.ncbi.nlm.nih.gov/geo/) under accession number GSE217518. Codes used for the analysis in this study have been deposited at https://github.com/chienlinglin/modeling-UTR-variants-stability (copy archived at *Su, 2025*). Other databases used in the study: UCSC PhyloP; AREsite2; ATtRACT; Ensembl; Harmonized Cancer Datasets; and Taiwan Biobanks.

The following dataset was generated:

| Author(s) | Year | Dataset title | Dataset URL | Database and Identifier |
|---|---|---|---|---|
| Lin C, Wang Y, Su J, Chang Y, Yang C, Lin P, Kang Y, Hsieh Y, Huang Y | 2025 | Multiplexed Assays of Human Disease-relevant Mutations Reveal UTR Dimer Composition as a Major Determinant of RNA Stability | https://www.ncbi.nlm.nih.gov/geo/query/acc.cgi?acc=GSE217518 | NCBI Gene Expression Omnibus, GSE217518 |

*Continued on next page*

The following previously published datasets were used:

| Author(s) | Year | Dataset title | Dataset URL | Database and Identifier |
|---|---|---|---|---|
| Wu Q, Medina SG, Kushawah G, DeVore ML | 2019 | Translation affects mRNA stability in a codon dependent manner in human cells | https://www.ncbi.nlm.nih.gov/geo/query/acc.cgi?acc=GSE126523 | NCBI Gene Expression Omnibus, GSE126523 |
| Mueller MB, Jayaraj GG, Hartl FU | 2023 | Mechanisms of stop codon readthrough mitigation reveal principles of GCN1 mediated translational quality control | https://www.ncbi.nlm.nih.gov/geo/query/acc.cgi?acc=GSE214396 | NCBI Gene Expression Omnibus, GSE214396 |

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
