## [Editor Report · eLife Assessment]

This **valuable** study combines massively parallel reporter assays and regression analysis to identify sequence features in untranslated regions contributing to the stability of in vitro transcribed mRNA delivered to cells. The strength of evidence presented is **solid**, although some points about half-life measurements and the relevance of identified sequence features to native transcript stability will inform future discussion surrounding the present study. Taken together, the work will be of interest to a broad swath of colleagues studying post-transcriptional gene regulation and especially to those using massively parallel reporter assays.

---

## [Referee Report · Reviewer #1 (Public review)]

In the manuscript by Su et al., the authors present a massively parallel reporter assay (MPRA) measuring the stability of in vitro transcribed mRNAs carrying wild-type or mutant 5' or 3' UTRs transfected into two different human cell lines. The goal presented at the beginning of the manuscript was to screen for effects of disease-associated point mutations on the stability of the reporter RNAs carrying partial human 5' or 3' UTRs. However, the majority of the manuscript is dedicated to identifying sequence components underlying the differential stability of reporter constructs. This analysis showed that UA dinucleotides are the most predictive feature of RNA stability in both cell lines and both UTRs.

The effect of AU rich elements (AREs) on RNA stability is well established in multiple systems, and the present study confirms this general trend, but points out variability in the consequence of seemingly similar motifs on RNA stability. For example, the authors report that a long stretch of Us has extreme opposite effects on RNA stability depending on whether it is preceded by an A (strongly destabilizing) or followed by an A (strongly stabilizing). While the authors interpretation of a context-dependence of the effect is certainly well-founded, it seems counterintuitive that the preceding or following A would be the (only) determining factor. This points to a generally reductionist approach taken by the authors in the analysis of the data and in their attempt to dissect the contribution of "AU rich sequences" to RNA stability, with a general tendency to reduce the size and complexity of the features (e.g. to dinucleotides). While this certainly increases the statistical power of the analysis due to the number of occurrences of these motifs, it limits the interpretability of the results. How do UA dinucleotides per se contribute to destabilizing the RNA, both in 5' and 3' UTRs, but (according to limited data presented) not in coding sequences? What is the mechanism? RBPs binding to UA dinucleotide containing sequences are suggested to "mask" the destabilizing effect, thereby leading to a more stable RNA. Gain of UA dinucleotides is reported to have a destabilizing effect, but again no hypothesis is provided as to the underlying molecular mechanism. In addition to reducing the motif length to dinucleotides, the notion of "context dependence" is used in a very narrow sense.

The present MPRA measures the effect of UTR sequences in one specific reporter context and using one experimental approach (following the decay of in vitro transcribed and transfected RNAs). While this method certainly has its merits compared to other approaches, it also comes with some caveats: RNA is delivered naked, without bound RBPs and no nuclear history, e.g. of splicing (no EJCs), editing and modifications. Therefore, it remains to be seen whether UA dinucleotide frequency is a substantial factor in determining the half-lives of endogenous mRNAs.

The authors conclude their study with a meta-analysis of genes with increased UA dinucleotides in 5' and 3'UTRs, showing that specific functional groups are overrepresented among these genes. In addition, they provide evidence for an effect of disease-associated UTR mutations on endogenous RNA stability. While these elements link back to the original motivation of the study (screening for effects of point mutations in 5' and 3' UTRs), they provide only a limited amount of additional insights.

In summary, this manuscript presents an interesting addition to the long-standing attempts at dissecting the sequence basis of RNA stability in human cells. The analysis is in general comprehensive and sound; however, it remains unclear to what extent the findings can be generalized beyond the method and the experimental system used here.

Comments on revisions:

Parts of my original comments have been adequately addressed by the reviewers.

After reading the revised manuscript and the rebuttal, my main concern is related to the figure comparing the half-lives as measured in the two different cell lines that was included in the response to reviewer 2, but not in the revised manuscript. The complete lack of correlation between the half-lives of the 3'UTR library measured in the two cell lines is concerning. While variability and cell type-specific effects can be expected, some principles should be the same (such as the effect of UA dinucleotides that the authors report), leading to at least some correlation.

In addition, it is unclear to me why the half-lives measured for the two libraries in HEK cells are shifted (median ln(t 1/2)=6-7 for the 5'UTR library and ln(t 1/2)=4-4.5 for the 3'UTR library), but not in SH.

I feel that this figure contains important information that should be included in the final manuscript.

---

## [Referee Report · Reviewer #2 (Public review)]

Summary of goals:

Untranslated regions are key cis-regulatory elements that control mRNA stability, translation, and translocation. Through interactions with small RNAs and RNA binding proteins, UTRs form complex transcriptional circuitry that allows cells to fine-tune gene expression. Functional annotation of UTR variants has been very limited, and improvements could offer insights into disease relevant regulatory mechanisms. The goals were to advance our understanding of the determinants of UTR regulatory elements and characterize the effects of a set of "disease-relevant" UTR variants.

Strengths:

The use of a massively parallel reporter assay allowed for analysis of a substantial set (6,555 pairs) of 5' and 3' UTR fragments compiled from known disease associated variants. Two cell types were used.

The findings confirm previous work about the importance of AREs, which helps show validity and adds some detailed comparisons of specific AU-rich motif effects in these two cell types.

Using a Lasso regression, TA-dinucleotide content is identified as a strong regulator of RNA stability in a context dependent manner based on GC content and presence of RNA binding protein binding motifs. The findings have potential importance, drawing attention to a UTR feature that is not well characterized.

The use of complementary datasets, including from half-life analyses of RNAs and from random sequence library MRPA's, is a useful addition and supports several important findings. The finding the TA dinucleotides have explanatory power separate from (and in some cases interacting with) GC content is valuable.

The functional enrichment analysis suggests some new ideas about how UTRs may contribute to regulation of certain classes of genes.

Weaknesses:

In this section, original reviewer comments about the initial submission and the responses of the authors are listed together with new reviewer responses to the authors:

Reviewer original comment 1: It is difficult to understand how the calculations for half-life were performed. The sequencing approach measures the relative frequency of each sequence at each time point (less stable sequences become relatively less frequent after time 0, whereas more stable sequences become relatively more frequent after time 0). Since there is no discussion of whether the abundance of the transfected RNA population is referenced to some external standard (e.g., housekeeping RNAs), it is not clear how absolute (rather than relative) half-lives were determined.

Author response: [The authors showed the equations used to calculate half lives based on read counts.] They stated that "The absolute abundance was not required for the half-life calculation."

Reviewer response to authors: The methods section states that DESeq2 was used to normalize read counts. DESeq2 normalization assumes that levels of most RNAs are not different between samples. That assumption is not valid here, since RNAs in the library are introduced into cells at time 0 and all RNAs decrease over time. If DESeq2 is applied without modification to normalize across timepoints, normalized reads from less stable RNAs will decrease over time (as expected) but normalized reads from more stable RNAs will increase. Can the authors please clarify in the methods how the read counts were normalized to account for this issue?

Reviewer original comment 2: Fig. S1A and B are used to assess reproducibility. They show that read counts at a given time point correlate well across replicate experiments. However, this is not a good way to assess reproducibility or accuracy of the measurements of t1/2 are. (The major source of variability in read counts in these plots - especially at early time points - is likely starting abundance of each RNA sequence, not stability.) This creates concerns about how well the method is measuring t1/2. Also creating concern is the observation that many RNAs are associated with half-lives that are much longer than the time points analyzed in the study. For example, based upon Figure S1 and Table S1 correctly, the median t1/2 for the 5' UTR library in HEK cells appears to be >700 minutes. Given that RNA was collected at 30, 75, and 120 minutes, accurate measurements of RNAs with such long half lives would seem to be very difficult.

Author response: ... The calculation of the half-life involves first determining the decay constant λ, which represents a constant rate of decay. Since λ is a constant, it is possible to accurately calculate it without needing data over the entire decay range. Our experimental design considers this by selecting appropriate time points to ensure a reliable estimation of λ, and thus, the half-life. To determine the most suitable time points, we conducted preliminary experiments using RT-PCR. These experiments indicated that 30, 75, and 120 minutes provided an effective range for capturing the decay dynamics of the transcripts.

Reviewer response to author comments: Based on Fig. S1D, for 3' UTRs in both cell types and for 5' UTRs in SH-SY5Y cells, median t1/2 is in the range of ~30 to 90 minutes (corresponding to ln t1/2 = 3.5 to 4.5). Measuring RNAs at 30, 75, and 120 minutes would therefore be a good choice for these cases, However, median t1/2 in HEK cells appears to be ~600 minutes (corresponding to ln t1/2 ~6.4) for HEK cells. For t1/2 of 600 minutes, RNA levels at the final time point (120 minutes) would be 90% of the those at the first time point (30 minutes), which illustrates why the method would need to be able to reliably capture very small changes in RNA abundance to accurately measure t1/2 for transcripts with half-lives much longer than 120 minutes. As suggested in our original review, this concern could be addressed by showing the correlation of half-lives across replicates for the 5' and 3' UTR libraries in both cell types. Alternatively, the authors could show other measures of reproducibility for the half-life measurements across replicates. This requires no additional experimentation and can be done using the data from replicate runs shown in Fig. S1A and B. We remain concerned that for sequences with very long half-lives, extrapolating the half-life from small changes between 30 and 120 minutes will lead to imprecise measurements.

Reviewer original comment 3: There is no direct comparison of t1/2 between the two cell types studied for the full set of sequences studied. This would be helpful in understanding whether the regulatory effects of UTRs are generally similar across cell lines (as has been shown in some previous studies) or whether there are fundamental differences. The distribution of t1/2's is clearly quite different in the two cell lines, but it is important to know if this reflects generally slow RNA turnover in HEK cells or whether there are a large number of sequence-specific effects on stability between cell lines. A related issue is that it is not clear whether the relatively small number of significant variant effects detected in HEK cells versus SH-SY5Y cells is attributable to real biological differences between cell types or to technical issues (many fewer read counts and much longer half lives in HEK cells).

Author response: For both cell lines, we selected oligonucleotides with R2 > 0.5 and mean squared error (MSE) < 1 for analysis when estimating half-life (λ) by linear regression. This selection criterion was implemented to minimize the effect of experimental noise. After quality control, we selected common UTRs and compared the RNA half-lives of the two cell lines using a scatter plot. The figure below shows that RNA half-lives are quite different between the cell lines, with a moderate similarity observed in the 5' UTRs (R = 0.21), while the correlation in the 3' UTRs is non-significant. Despite the low correlation of mRNA half-life between the two cell lines, UA-dinucleotide and UA-rich sequences consistently emerge as the most significant destabilizing features, suggesting a shared regulatory mechanism across diverse cellular environments.

Reviewer response to author comments: We appreciate that the authors shared this additional analysis of the data. We believe that this is an important finding and that the additional figure showing correlations of half-lives across cell types should be included in the manuscript or supplement. Discussion of this result in the manuscript would also be useful for readers. This result is surprising to us since we would have expected that widely expressed RNA-binding proteins would have led to more similar effects between the two cell types, as previously found using other approaches (e.g., studies of 3' UTR effects in MPRAs). It would also be appropriate to discuss that differences seen between the two cell types indicate that caution is warranted when trying to generalize the results of this study to other cell types.

Reviewer original comment 4 has been addressed adequately in the revised manuscript.

Appraisal and impact:

Reviewer original comment 1: The work adds to existing studies that previously identified sequence features, including AREs and other RNA binding protein motifs, that regulate stability and puts a new emphasis on the role of "TA" (better "UA") dinucleotides. It is not clear how potential problems with the RNA stability measurements discussed above might influence the overall conclusions, which may limit the impact unless these can be addressed.

It is difficult to understand whether the importance of TA dinucleotides is best explained by their occurrence in a related set of longer RBP binding motifs (see Fig 5J, these motifs may be encompassed by the "WWWWWW cluster") or whether some other explanation applies. Further discussion of this would be helpful. Does the LASSO method tend to collapse a more diverse set of longer motifs that are each relatively rare compared to the dinucleotide? It remains unclear whether TA dinucleotides are associated with less stability independent of the presence of the known larger WWWWWWW motif. As noted above, the importance of TA dinucleotides in the HEK experiments appears to be less than is implied in the text.

Author response: To ensure the representativeness of the features entered into the LASSO model, we pre-selected those with an occurrence greater than 10% among all UTRs. There is no evidence to support a preference for dinucleotides by LASSO. To address whether the destabilizing effect of UA dinucleotides is part of the broader WWWWWW motif, we divided UA dinucleotides into two groups: those within the WWWWWW motif and those outside of it. Specifically, we divided UTRs into two categories: 'at least one UA within a WWWWWW motif' and 'no UA within a WWWWWW motif,' and visualized the results using a boxplot. As shown in [figures provided to the reviewers], the destabilizing trend still remains for UA dinucleotides outside of the WWWWWW motif, although the effect appears to be more pronounced when UA is within the WWWWWW motif. This suggests that while UA dinucleotides have a destabilizing effect independently, their impact is amplified when they are part of the broader WWWWWW motif.

Reviewer response to authors: These are useful additional analyses, and we suggest that the additional figure and discussion should be included in the manuscript/supplement so that readers can benefit from them.

Reviewer original comment 2: The inclusion of more than a single cell type is an acknowledgement of the importance of evaluating cell type-specific effects. The work suggests a number of cell type-specific differences, but due to technical issues (especially with the HEK data, as outlined above) and the use of only two cell lines, it is difficult to understand cell type effects from the work.

The inclusion of both 3' and 5' UTR sequences distinguishes this work from most prior studies in the field. Contrasting the effects of these regions on stability is of interest, although the role of these UTRs (especially the 5' UTR) in translational regulation is not assessed here.

Author response: We examined the role of UTR and UTR variants in translation regulation using polysome profiling. By both univariate analysis and an elastic regression model, we identified motifs of short repeated sequences, including SRSF2 binding sites, as mutation hotspots that lead to aberrant translation. Furthermore, these polysome-shifting mutations had a considerable impact on RNA secondary structures, particularly in upstream AUG-containing 5' UTRs. Integrating these features, our model achieved high accuracy (AUROC > 0.8) in predicting polysome-shifting mutations in the test dataset. Additionally, metagene analysis indicated that pathogenic variants were enriched at the upstream open reading frame (uORF) translation start site, suggesting changes in uORF usage underlie the translation deficiencies caused by these mutations. Illustrating this, we demonstrated that a pathogenic mutation in the IRF6 5' UTR suppresses translation of the primary open reading frame by creating a uORF. Remarkably, site-directed ADAR editing of the mutant mRNA rescued this translation deficiency. Because the regulation of translation and stability does not converge, we illustrate these two mechanisms in two separate manuscripts (this one and doi.org/10.1101/2024.04.11.589132).

Reviewer response to authors: This is useful context. No further comment.

---

## [Referee Report · Reviewer #3 (Public review)]

Summary:

In their manuscript titled "Multiplexed Assays of Human Disease‐relevant Mutations Reveal UTR Dinucleotide Composition as a Major Determinant of RNA Stability" the authors aim to investigate the effect of sequence variations in 3'UTR and 5'UTRs on the stability of mRNAs in two different human cell lines.

To do so, the authors use a massively parallel reporter assay (MPRA). They transfect cells with a set of mRNA reporters that contain sequence variants in their 3' or 5' UTRs, which were previously reported in human diseases. They follow their clearance from cells over time relative to the matching non-variant sequence. To analyze their results, they define a set of factors (RBP and miRNA binding sites, sequence features, secondary structure etc.) and test their association with differences in mRNA stability. For features with a significant association, they use clustering to select a subset of factors for LASSO regression and identify factors that affect mRNA stability.

They conclude that the TA dinucleotide content of UTRs is the strongest destabilizing sequence feature. Within that context, elevated GC content and protein binding can protect susceptible mRNAs from degradation. They also show that TA dinucleotide content of UTRs affects native mRNA stability and that it is associated with specific functional groups. Finally, they link disease associated sequence variants with differences in mRNA stability of reporters.

Strengths:

(1) This work introduces a different MPRA approach to analyze the effect of genetic variants. While previous works in tissue culture use DNA transfections that require normalization for transcription efficiency, here the mRNA is directly introduced into cells at fixed amounts, allowing a more direct view of the mRNA regulation.

(2) The authors also introduce a unique analysis approach, which takes into account multiple factors that might affect mRNA stability. This approach allows them to identify general sequence features that affect mRNA stability beyond specific genetic variants, and reach important insights on mRNA stability regulation. Indeed, while the conclusions to genetic variants identified in this work are interesting, the main strength of the work involves general effect of sequence features rather than specific variants.

(3) The authors provide adequate support for their claims and validate their analysis using both their reporter data and native genes. For the main feature identified, TA di-nucleotides, they perform follow-up experiments with modified reporters that further strengthen their claims, and also validate the effect on native cellular transcripts (beyond reporters), demonstrating its validity also within native scenarios.

(4) The work provides a broad analysis of mRNA stability, across two mRNA regulatory segments (3'UTR and 5'UTR) and is performed in two separate cell-types. Comparison between two different cell-types is adequate, and the results demonstrate, as expected, the dependence of mRNA stability on the cellular context. Analysis of 3'UTR and 5'UTR regulatory effects also shows interesting differences and similarities between these two regulatory regions.

Weaknesses:

In their revised manuscripts, the authors successfully address many of the weaknesses raised in the original review, including the effect of possible confounding effects, and additional methodology details. Notably, two of the issues raised in the original report, have only been partially addressed in the revision.

(1) The analysis and regression models built in this work are not thoroughly investigated relative to native genes within cells.

While using MPRAs indeed allows to isolate regulatory effects that are less influential in-vivo, the resulting effects still provide some regulatory function in-vivo. The goal of such an analysis would not be to demonstrate the predictive power of the models, or to make any claims regarding using these models to fully explain or predict the stability of native transcripts. Clearly, additional more prominent factors could function in controlling endogenous RNA stability.

Instead, the goal of such an investigation is to simply assess the fraction of in-vivo regulation that the factors identified in this work contribute in native contexts, and what is the relative contribution of the phenomena captured by the well-controlled MPRA study.

This reviewer believes that even if the effects identified by the current MPRA study only contribute a small fraction of in-vivo variation, an analysis that aim to estimate what this fraction is, will be very relevant to this study for several reasons. First, in order to appreciate the results of this study within their in-vivo context. Second, in light of the questions raised as motivation for this study, and particularly the need to identify the effect of disease-associated 3'UTR variants, which clearly have an in-vivo effect.

(2) Methodology validation can be performed with simulated data (generated in-silico by the authors) to provide an independent support for the ability of the current methodology to correctly extract regulatory effects from the data.

---

## [Author Response]

The following is the authors’ response to the original reviews.

**Public Reviews:**

**Reviewer #1 (Public Review):**
In the manuscript by Su et al., the authors present a massively parallel reporter assay (MPRA) measuring the stability of in vitro transcribed mRNAs carrying wild-type or mutant 5' or 3' UTRs transfected into two different human cell lines. The goal presented at the beginning of the manuscript was to screen for effects of disease-associated point mutations on the stability of the reporter RNAs carrying partial human 5' or 3' UTRs. However, the majority of the manuscript is dedicated to identifying sequence components underlying the differential stability of reporter constructs. This shows that TA dinucleotides are the most predictive feature of RNA stability in both cell lines and both UTRs.The effect of AU rich elements (AREs) on RNA stability is well established in multiple systems, and the present study confirms this general trend but points out variability in the consequence of seemingly similar motifs on RNA stability. For example, the authors report that a long stretch of Us has extreme opposite effects on RNA stability depending on whether it is preceded by an A (strongly destabilizing) or followed by an A (strongly stabilizing). While the authors interpretation of a context- dependence of the effect is certainly well-founded, it seems counterintuitive that the preceding or following A would be the (only) determining factor. This points to a generally reductionist approach taken by the authors in the analysis of the data and in their attempt to dissect the contribution of "AU rich sequences" to RNA stability, with a general tendency to reduce the size and complexity of the features (e.g. to dinucleotides). While this certainly increases the statistical power of the analysis due to the number of occurrences of these motifs, it limits the interpretability of the results. How do TA dinucleotides per se contribute to destabilizing the RNA, both in 5' and 3' UTRs, but (according to limited data presented) not in coding sequences? What is the mechanism? RBPs binding to TA dinucleotide containing sequences are suggested to "mask" the destabilizing effect, thereby leading to a more stable RNA. Gain of TA dinucleotides is reported to have a destabilizing effect, but again no hypothesis is provided as to the underlying molecular mechanism. In addition to reducing the motif length to dinucleotides, the notion of "context dependence" is used in a very narrow sense; especially when focusing on simple and short motifs, a more extensive analysis of the interdependence of these features (beyond the existing analysis of the relationship between TA- diNTs and GC content) could potentially reveal more of the context dependence underlying the seemingly opposite behavior of very similar motifs.

(We have used UA instead of TA, as per the reviewer's suggestion)

The contribution of coding region sequence to RNA stability has been extensively discussed (For example: doi.org/10.1016/j.molcel.2022.03.032; doi.org/10.1186/s13059-020-02251-5; doi.org/10.15252/embr.201948220; doi.org/10.1371/journal.pone.0228730; doi.org/10.7554/eLife.45396). While UA content at the third codon position (wobble position) has been implicated as a pro-degradation signal, codon optimality has emerged as the most prominent determinant for RNA stability. This indicates that the role of coding regions in RNA stability differs from that of UTRs due to the involvement of translation elongation. We did not intend to suggest that UA-dinucleotides in UTRs and coding regions have the same effect.

To ensure the representativeness of the features entered into the LASSO model, we pre-selected those with an occurrence greater than 10% among all UTRs. As a result, while motifs with very low occurrences were excluded from the analysis, there is no evidence to indicate a preference for dinucleotides by the LASSO model.

We hypothesize that UA-dinucleotide may recruit endonucleases RNase A family, whose catalytic pockets exhibit a strong bias for UA dinucleotide (doi.org/10.1016/j.febslet.2010.04.018). Structures or protein bindings that block this recognition might stabilize RNAs. To gain further insight into the motif interactions, we investigated the interactions between UA and other 15 dinucleotides through more detailed analyses. We conducted a linear regression analysis investigating interactions between UA and the other 15 dinucleotides. The formula used below includes UA:

ln⁡(t1/2)i=β0+βUAXUAi+βDiNTXDiNTi+βGC%XGC%i
+βUAXGCXUAiXGC%i
+βUAXDiNTXUAiXDiNT)
+ϵi, where all *β* terms represent the regression coefficients, and XUAi, XDiNT, and ^th^ UTR, respectively, and 𝜖_i_ denotes the error term. For each dinucleotide, we tested the significance of *β*
XGC% represent the number of UA dinucleotides, the number of other dinucleotides (other than UA), and the GC content of the i*UAxGC%* and *βUAxDiNT*, and compared their p-values using a quantile-quantile (QQ) plot. Author response image 1 shows that the interaction effect of UA dinucleotides with GC% is much more significant than interactions with the other 15 dinucleotides, as indicated by the inflated QQ plot of p-values. This suggests that GC content is a more critical contextual factor influencing UA dinucleotides' impact on RNA stability.

**Author response image 1. sa4fig1:** 

The present MPRAs measures the effect of UTR sequences in one specific reporter context and using one experimental approach (following the decay of in vitro transcribed and transfected RNAs). While this approach certainly has its merits compared to other approaches, it also comes with some caveats: RNA is delivered naked, without bound RBPs and no nuclear history, e.g. of splicing (no EJCs), editing and modifications. One way to assess the generalizability of the results as well as the context dependence of the effects is to perform the same analysis on existing datasets of RNA stability measurements obtained through other methods (e.g. transcription inhibition). Are TA dinucleotides universally the most predictive feature of RNA half-lives?

Our system studies the stability control of RNA synthesized in vitro and delivered into human cells. While we did not intend to generalize our conclusions to endogenous RNAs, our approach contributes to the understanding of in vitro synthesized RNA used for cellular expression, such as in vaccines. It is known that endogenous RNAs undergo very different regulation. The most prominent factors controlling endogenous RNA stability are the density of splice junctions and the length of UTRs (doi.org/10.1186/s13059-022-02811-x; doi.org/10.1186/s12915-021-00949-x). To decipher the sequence regulation, these factors are controlled in our experiments. Therefore, we do not expect the dinucleotide features found by our approach to be generalized as the most predictive feature of RNA half-life in vivo.

The authors conclude their study with a meta-analysis of genes with increased TA dinucleotides in 5' and 3'UTRs, showing that specific functional groups are overrepresented among these genes. In addition, they provide evidence for an effect of disease-associated UTR mutations on endogenous RNA stability. While these elements link back to the original motivation of the study (screening for effects of point mutations in 5' and 3' UTRs), they provide only a limited amount of additional insights.

We utilized the Taiwan Biobank to investigate whether mutations significantly affecting RNA stability also impact human biochemical measurements. Our findings indicate that these mutations indeed have a significant effect on various biochemical indices. This highlights the importance of our study, as it bridges basic science with potential applications in precision medicine. By linking specific UTR mutations with measurable changes in biochemical indices, our research underscores the potential for these findings to inform targeted medical interventions in the future.

In summary, this manuscript presents an interesting addition to the long-standing attempts at dissecting the sequence basis of RNA stability in human cells. The analysis is in general very comprehensive and sound; however, at times the goal of the authors to find novelty and specificity in the data overshadows some analyses. One example is the case where the authors try to show that TA-dinucleotides and GC content are decoupled and not merely two sides of the same coin.They claim that the effect of TA dinucleotides is different between high- and low-GC content contexts but do not control for the fact that low GC-content regions naturally will contain more TA dinucleotides and therefore the effect sizes and the resulting correlation between TA-diNT rate and stability will be stronger (Fig. 5A). A more thorough analysis and greater caution in some of the claims could further improve the credibility of the conclusions.

Low GC content implies a higher UA content but does not directly equate to a high UA-dinucleotide ratio. For instance, the sequence AUUGAACCUU has a lower GC content (0.3) compared to UAUAGGCCGC (0.6), yet it also has a lower UA-dinucleotide ratio (0 vs. 0.22). To address this concern more rigorously, we performed a stratified analysis based on UA-diNT rate. As shown in our Fig. S7C, even after stratifying by UA- dinucleotide ratio (upper panel high UA- dinucleotide ratio / lower panel low UA- dinucleotide ratio), we still observe that the destabilizing effect of UA is stronger in the low GC content group.

**Reviewer #2 (Public Review):**
Summary of goals:Untranslated regions are key cis-regulatory elements that control mRNA stability, translation, and translocation. Through interactions with small RNAs and RNA binding proteins, UTRs form complex transcriptional circuitry that allows cells to fine-tune gene expression. Functional annotation of UTR variants has been very limited, and improvements could offer insights into disease relevant regulatory mechanisms. The goals were to advance our understanding of the determinants of UTR regulatory elements and characterize the effects of a set of "disease-relevant" UTR variants.Strengths:The use of a massively parallel reporter assay allowed for analysis of a substantial set (6,555 pairs) of 5' and 3' UTR fragments compiled from known disease associated variants. Two cell types were used.The findings confirm previous work about the importance of AREs, which helps show validity and adds some detailed comparisons of specific AU-rich motif effects in these two cell types.Using a Lasso regression, TA-dinucleotide content is identified as a strong regulator of RNA stability in a context dependent manner based on GC content and presence of RNA binding protein binding motifs. The findings have potential importance, drawing attention to a UTR feature that is not well characterized.The use of complementary datasets, including from half-life analyses of RNAs and from random sequence library MRPA's, is a useful addition and supports several important findings. The finding the TA dinucleotides have explanatory power separate from (and in some cases interacting with) GC content is valuable.The functional enrichment analysis suggests some new ideas about how UTRs may contribute to regulation of certain classes of genes.Weaknesses:It is difficult to understand how the calculations for half-life were performed. The sequencing approach measures the relative frequency of each sequence at each time point (less stable sequences become relatively less frequent after time 0, whereas more stable sequences become relatively more frequent after time 0). Since there is no discussion of whether the abundance of the transfected RNA population is referenced to some external standard (e.g., housekeeping RNAs), it is not clear how absolute (rather than relative) half-lives were determined.

We estimated decay constant λ and half-life (*t*_1/2_) by the following equations:ln⁡(Ci(t)Ci(t=0))=−λtit1/2=ln⁡(2)λ

where *C*_i(t)_ and *C*_i(t=0)_ are read count values of the ith replicate at time points *t* and 0 (see also Methods). The absolute abundance was not required for the half-life calculation.

Fig. S1A and B are used to assess reproducibility. They show that read counts at a given time point correlate well across replicate experiments. However, this is not a good way to assess reproducibility or accuracy of the measurements of t1/2 are. (The major source of variability in read counts in these plots - especially at early time points - is likely the starting abundance of each RNA sequence, not stability.) This creates concerns about how well the method is measuring t1/2. Also creating concern is the observation that many RNAs are associated with half-lives that are much longer than the time points analyzed in the study. For example, based upon Figure S1 and Table S1 correctly, the median t1/2 for the 5' UTR library in HEK cells appears to be >700 minutes. Given that RNA was collected at 30, 75, and 120 minutes, accurate measurements of RNAs with such long half lives would seem to be very difficult.

We estimated the half-life based on the following equations:ln⁡(Ci(t)Ci(t=0))=−λtit1/2=ln⁡(2)λ

where *C*_i(t)_ and *C*_i(t=0)_ are read count values of the ith replicate at time points *t* and 0 (see also Methods). The calculation of the half-life involves first determining the decay constant λ, which represents a constant rate of decay. Since λ is a constant, it is possible to accurately calculate it without needing data over the entire decay range. Our experimental design considers this by selecting appropriate time points to ensure a reliable estimation of λ, and thus, the half-life. To determine the most suitable time points, we conducted preliminary experiments using RT-PCR.

These experiments indicated that 30, 75, and 120 minutes provided an effective range for capturing the decay dynamics of the transcripts.

There is no direct comparison of t1/2 between the two cell types studied for the full set of sequences studied. This would be helpful in understanding whether the regulatory effects of UTRs are generally similar across cell lines (as has been shown in some previous studies) or whether there are fundamental differences. The distribution of t1/2's is clearly quite different in the two cell lines, but it is important to know if this reflects generally slow RNA turnover in HEK cells or whether there are a large number of sequence-specific effects on stability between cell lines. A related issue is that it is not clear whether the relatively small number of significant variant effects detected in HEK cells versus SH-SY5Y cells is attributable to real biological differences between cell types or to technical issues (many fewer read counts and much longer half lives in HEK cells).

For both cell lines, we selected oligonucleotides with R^2^ > 0.5 and mean squared error (MSE) < 1 for analysis when estimating half-life (λ) by linear regression. This selection criterion was implemented to minimize the effect of experimental noise. After quality control, we selected common UTRs and compared the RNA half-lives of the two cell lines using a scatter plot. Author response image 2 shows that RNA half-lives are quite different between the cell lines, with a moderate similarity observed in the 5' UTRs (R = 0.21), while the correlation in the 3' UTRs is non-significant.

**Author response image 2. sa4fig2:** 

Despite the low correlation of mRNA half-life between the two cell lines, UA-dinucleotide and UA-rich sequences consistently emerge as the most significant destabilizing features, suggesting a shared regulatory mechanism across diverse cellular environments.

The general assertion is made in many places that TA dinucleotides are the most prominent destabilizing element in UTRs (e.g., in the title, the abstract, Fig. 4 legend, and on p. 12). This appears to be true for only one of the two cell lines tested based on Fig. 3.

UA-dinucleotides and other UA-rich sequences exhibit similar effects on RNA stability, as illustrated in Fig. S5A-C. In two cell lines, UA-dinucleotide and WWWWWW sequences were representatives of the same stability-affecting cluster. While the impact of UA-dinucleotides can be generalized, we have rephrased some statements for clarification to avoid any potential misunderstanding. For examples:

Abstract: “...We found that UA dinucleotides and UA-rich motifs are the most prominent destabilizing element.“

p.10: “UA dinucleotides and UA-rich motifs are the most common and effective RNA destabilizing factor”

Figure 4: “The UTR UA dinucleotides and UA-rich motifs are the most common and influential RNA destabilizing factor.”

Appraisal and impact:The work adds to existing studies that previously identified sequence features, including AREs and other RNA binding protein motifs, that regulate stability and puts a new emphasis on the role of "TA" (better "UA") dinucleotides. It is not clear how potential problems with the RNA stability measurements discussed above might influence the overall conclusions, which may limit the impact unless these can be addressed.It is difficult to understand whether the importance of TA dinucleotides is best explained by their occurrence in a related set of longer RBP binding motifs (see Fig 5J, these motifs may be encompassed by the "WWWWWW cluster") or whether some other explanation applies. Further discussion of this would be helpful. Does the LASSO method tend to collapse a more diverse set of longer motifs that are each relatively rare compared to the dinucleotide? It remains unclear whether TA dinucleotides are associated with less stability independent of the presence of the known larger WWWWWWW motif. As noted above, the importance of TA dinucleotides in the HEK experiments appears to be less than is implied in the text.

To ensure the representativeness of the features entered into the LASSO model, we pre-selected those with an occurrence greater than 10% among all UTRs. There is no evidence to support a preference for dinucleotides by LASSO. To address whether the destabilizing effect of UA dinucleotides is part of the broader WWWWWW motif, we divided UA dinucleotides into two groups: those within the WWWWWW motif and those outside of it. Specifically, we divided UTRs into two categories: 'at least one UA within a WWWWWW motif' and 'no UA within a WWWWWW motif,' and visualized the results using a boxplot. As shown in Author response image 3, the destabilizing trend still remains for UA dinucleotides outside of the WWWWWW motif, although the effect appears to be more pronounced when UA is within the WWWWWW motif. This suggests that while UA dinucleotides have a destabilizing effect independently, their impact is amplified when they are part of the broader WWWWWW motif.

**Author response image 3. sa4fig3:** 

The inclusion of more than a single cell type is an acknowledgement of the importance of evaluating cell type-specific effects. The work suggests a number of cell type-specific differences, but due to technical issues (especially with the HEK data, as outlined above) and the use of only two cell lines, it is difficult to understand cell type effects from the work.The inclusion of both 3' and 5' UTR sequences distinguishes this work from most prior studies in the field. Contrasting the effects of these regions on stability is of interest, although the role of these UTRs (especially the 5' UTR) in translational regulation is not assessed here.

We examined the role of UTR and UTR variants in translation regulation using polysome profiling. By both univariate analysis and an elastic regression model, we identified motifs of short repeated sequences, including SRSF2 binding sites, as mutation hotspots that lead to aberrant translation. Furthermore, these polysome-shifting mutations had a considerable impact on RNA secondary structures, particularly in upstream AUG-containing 5’ UTRs. Integrating these features, our model achieved high accuracy (AUROC > 0.8) in predicting polysome-shifting mutations in the test dataset. Additionally, metagene analysis indicated that pathogenic variants were enriched at the upstream open reading frame (uORF) translation start site, suggesting changes in uORF usage underlie the translation deficiencies caused by these mutations. Illustrating this, we demonstrated that a pathogenic mutation in the IRF6 5’ UTR suppresses translation of the primary open reading frame by creating a uORF. Remarkably, site-directed ADAR editing of the mutant mRNA rescued this translation deficiency. Because the regulation of translation and stability does not converge, we illustrate these two mechanisms in two separate manuscripts (this one and doi.org/10.1101/2024.04.11.589132).

**Reviewer #3 (Public Review):**
Summary:In their manuscript titled "Multiplexed Assays of Human Disease‐relevant Mutations Reveal UTRDinucleotide Composition as a Major Determinant of RNA Stability" the authors aim to investigate the effect of sequence variations in 3'UTR and 5'UTRs on the stability of mRNAs in two different human cell lines.To do so, the authors use a massively parallel reporter assay (MPRA). They transfect cells with a set of mRNA reporters that contain sequence variants in their 3' or 5' UTRs, which were previously reported in human diseases. They follow their clearance from cells over time relative to the matching non-variant sequence. To analyze their results, they define a set of factors (RBP and miRNA binding sites, sequence features, secondary structure etc.) and test their association with differences in mRNA stability. For features with a significant association, they use clustering to select a subset of factors for LASSO regression and identify factors that affect mRNA stability.They conclude that the TA dinucleotide content of UTRs is the strongest destabilizing sequence feature. Within that context, elevated GC content and protein binding can protect susceptible mRNAs from degradation. They also show that TA dinucleotide content of UTRs affects native mRNA stability, and that it is associated with specific functional groups. Finally, they link disease associated sequence variants with differences in mRNA stability of reporters.Strengths:This work introduces a different MPRA approach to analyze the effect of genetic variants. While previous works in tissue culture use DNA transfections that require normalization for transcription efficiency, here the mRNA is directly introduced into cells at fixed amounts, allowing a more direct view of the mRNA regulation.The authors also introduce a unique analysis approach, which takes into account multiple factors that might affect mRNA stability. This approach allows them to identify general sequence features that affect mRNA stability beyond specific genetic variants, and reach important insights on mRNA stability regulation. Indeed, while the conclusions to genetic variants identified in this work are interesting, the main strength of the work involve general effect of sequence features rather than specific variants.The authors provide adequate supports for their claims, and validate their analysis using both their reporter data and native genes. For the main feature identified, TA di-nucleotides, they perform follow-up experiments with modified reporters that further strengthen their claims, and also validate the effect on native cellular transcripts (beyond reporters), demonstrating its validity also within native scenarios.The work provides a broad analysis of mRNA stability, across two mRNA regulatory segments (3'UTR and 5'UTR) and is performed in two separate cell-types. Comparison between two different cell-types is adequate, and the results demonstrate, as expected, the dependence of mRNA stability on the cellular context. Analysis of 3'UTR and 5'UTR regulatory effects also shows interesting differences and similarities between these two regulatory regions.Weaknesses:(1) The authors fail to acknowledge several possible confounding factors of their MPRA approach in the discussion.First, while transfection of mRNA directly into cells allows to avoid the need to normalize for differences in transcription, the introduction of naked mRNA molecules is different than native cellular mRNAs and could introduce biases due to differences in mRNA modifications, protein associations etc. that may occur co-transcriptionally.Second, along those lines, the authors also use in-vitro polyadenylation. The length of the polyA tail of the transfected transcripts could potentially be very different than that of native mRNAs and also affect stability.

The transcripts used in our study were polyadenylated in vitro with approximately 100 nucleotides

(Fig. S1C), similar to the polyA tail lengths typically observed in vivo (dx.doi.org/10.1016/j.molcel.2014.02.007). Additionally, these transcripts were capped to emulate essential mRNA characteristics and to minimize immune responses in recipient cells. This design allows us to study RNA decay for in vitro-synthesized RNA delivered into human cells, akin to RNA vaccines, but it does not necessarily extend to endogenous RNAs. As mentioned, endogenous RNAs undergo nuclear processing and are decorated by numerous trans factors, resulting in distinct regulatory mechanisms. We therefore provided a more discussion on these differences and their implications in the revised manuscript: “However, while our approach effectively assesses the stability of synthesized RNA in human cells, it may not fully capture the decay dynamics of nuclear-synthesized RNA, which can be influenced by endogenous modifications and trans-acting RNA binding factors. (p. 18)”

(2) The analysis approach used in this work for identifying regulatory features in UTRs was not previously used. As such, lack of in-depth details of the methodology, and possibly also more general validation of the approach, is a drawback in convincing the reader in the validity of this approach and its results.In particular, a main point that is not addressed is how the authors decide on the set of "factors" used in their analysis? As choosing different sets of factors might affect the results of the analysis.

In our study, we employed the calculation of the Variance Inflation Factor (VIF) as a basis for selecting variables. This well-established method is widely used to detect variables with high collinearity, thus ensuring the robustness and reliability of our analysis. By identifying and excluding highly collinear variables, we aimed to minimize multicollinearity and improve the accuracy of our regression models. For more detailed information on the use of VIF in regression analysis, please refer to Akinwande, M., Dikko, H., and Samson, A. (2015). Variance Inflation Factor: As a Condition for the Inclusion of Suppressor Variable(s) in Regression Analysis. Open Journal of Statistics, 5, 754-767. doi: 10.4236/ojs.2015.57075. We have included the method details in the revised manuscript (p. 28) :”… to avoid multicollinearity caused by similar features that perturb feature selection, all features were clustered using single-linkage hierarchical clustering with the distance metric defined as one minus the absolute value of the Spearman correlation coefficient. We cut the tree at a specific height, and the feature that had the greatest influence on RNA stability, which was examined using a simple linear regression model, was selected to be the representative of each cluster. Then we calculated the variance inflation factor (VIF) value of the representative features. The VIFs were obtained by the following linear model and equations:X^ij=α^0(j)+∑k≠jα^k(j)XikRj2=∑i(X^ij−X¯⋅j)2∑i(Xij−X¯⋅j)2VIFj=11−Rj2

where X^ij and Xik are the estimated value of the jth feature and the value of the kth feature of the ith UTR (note that the kth feature is a feature other than the jth feature), α^0(j) and α^k(j) are the intercept and the regression coefficients of the linear model that regressed the jth feature on the other remaining features, and X_⋅j_ is the mean level of the jth feature of all UTRs.”

For example, the choice to use 7-mer sequences within the factors set is not explained, particularly when almost all motifs that are eventually identified (Figure 3B-E) are shorter.

The known RBP motifs are primarily 6-mer. To explore the possibility of discovering novel motifs that could significantly impact our model, we started with 7-mer sequences. However, our analysis revealed that including these additional variables did not improve the explanatory power of the model; instead, it reduced it. Consequently, our final model focuses on motifs shorter than 7-mer. We explained the motif selections in the revised manuscript (p. 9): “Given our discovery that the effect of AREs is heavily dependent on sequence content, we decided to further explore the effects of other sequence elements, i.e., beyond known regulatory motifs, in more detail. Since most reported RBP motifs are 6-mers, we initiated a search for novel motifs by analyzing the presence of all 7-mers in our massively parallel reporter assay (MPRA) library, correlating their occurrence with mRNA half-life.”

In addition, the authors do not perform validations to demonstrate the validity of their approach on simulated data or well-established control datasets. Such analysis would be helpful to further convince the reader in the usefulness and robustness of the analysis.

We acknowledge the importance of validating our approach on simulated data or well-established control datasets to demonstrate its robustness and reliability. However, to the best of our knowledge, there are currently no well-established control datasets available that perfectly correspond to our specific study context. Despite this, we will continue to search for any relevant datasets that could be utilized for this purpose in future work. This effort will help to further reinforce the confidence in our methodology and its findings.

(3) The analysis and regression models built in this work are not thoroughly investigated relative to native genes within cells. The effect of sequence "factors" on native cellular transcripts' stability is not investigated beyond TA di-nucleotides, and it is unclear to what degree do other predicted factors also affect native transcripts.

Our system studies the stability control of RNA synthesized in vitro and delivered into human cells. While we validated the UTR UA-dinucleotide effect in vivo, we did not intend to conclude that this is the most influential regulation for endogenous RNAs. It is known that endogenous RNAs undergo very different regulation. The most prominent factors controlling endogenous RNA stability are the density of splice junctions and the length of UTRs (doi.org/10.1186/s13059-022-02811-x; doi.org/10.1186/s12915-021-00949-x). To decipher the sequence regulation, we controlled for these factors in our experiments. Therefore, we acknowledge that several endogenous features, which were excluded by our approach, may serve as predictive features of RNA half-life in vivo.

**Recommendations for the authors:**

**Reviewer #1 (Recommendations For The Authors):**
Specific comments:Some references are missing, e.g for the sentence:

Please see the response below.

"Similarly, point mutation of the GFPT1 3' UTR results in congenital myasthenic syndrome." (p5)

The reference has been added to the text:

Dusl, M., Senderek, J., Muller, J. S., Vogel, J. G., Pertl, A., Stucka, R., Lochmuller, H., David, R., & Abicht, A. (2015). A 3'-UTR mutation creates a microRNA target site in the GFPT1 gene of patients with congenital myasthenic syndrome. Human Molecular Genetics, 24(12), 34183426. https://doi.org/10.1093/hmg/ddv090

"...but there have been no systematic assessments of the explicit effects of variants of both UTRs on stability regulation." (not true in the current phrasing; e.g. PMIDs 32719458, 36156153, 34849835)

These references have been added to the text. However, we have to point out that these studies do not focus on the effects of the disease-relevant variants. To clarify, we modified the sentence to "... systematic assessments of the explicit effects of disease-relevant variants in both UTRs on stability regulation are still absent."

"Multiple approaches have revealed AREs as exerting a destabilizing effect on RNA stability (Barreau et al., 2005). (p8)

The reference has been added to the text:

Barreau, C., Paillard, L., & Osborne, H. B. (2005). AU-rich elements and associated factors: are there unifying principles? Nucleic Acids Research, 33(22), 7138-7150. https://doi.org/10.1093/nar/gki1012

"This effect is specific, as such ratios in the coding region are inconsequential." (p12)

This refers to our findings of Fig. 4G and Supplemental Fig. S5F.

What are the sequences at the 5' and 3'UTR without insertion of a library? 5'UTR library (especially in SH) has much longer half-life compared to 3'utr library (Fig S1D).

There is no designed 5’UTR of the 3’UTR library, only the Kozak sequence derived from the pEGFPC1 vector. This may partially underlie the shorter half-life of the 3’ UTR library.

Fig2A: What are the units? "half-life (log)" Do the numbers correspond to log10(min)?

It represents ln (min). To clarify, we now use ‘ln t_1/2_ (min)’ in all figures.

Fig 2 and 3: This was done only on the wild-type sequences? Or all tested sequences together, wt and mut?

It was done only on the wild-type sequences. To clarify, we modified the text to “we examined the effect of AREs on RNA stability of the ref alleles according to specific sequence content….(p.8)” and “We considered as many factors as possible to explain the half-life of our ref UTR libraries,…. (p.9)”. ‘ref’ stands for reference.

"Furthermore, to avoid collinearity confounding our model, e.g., the effects of very similar factors (such as 'AA' and 'AAA' sequences), we clustered the factors according to their properties, and then only one representative factor from within a cluster (i.e., the one with the highest correlation to halflife within a cluster) was subjected to LASSO regression": Given the observed context dependence, e.g. in the case of poly-U stretches: Isn't this clustering leading to similar/identical motifs with different context being grouped together such as polyU preceded by an A (strongly destabilizing, according to Fig 2B) or followed by one (strongly stabilizing, according to Fig 2B), resulting in ignoring the context or using one potential outcome while a motif from the same cluster can have the opposite effect?

Thank you very much for pointing this out. To determine if considering different contextual effects within each feature cluster would enhance model performance, we modified our feature selection by choosing both the feature with the largest positive and the largest negative effect on RNA half-life in Step III of Figure 3A. We then split the data into a 2:1 training and testing set and repeated this process 100 times. Model performance was evaluated using mean average error (MAE), root mean squared error (RMSE), and adjusted R-squared. From Author response image 4, we observed no significant improvement in model performance using this new approach. Notably, in the SH-SY5Y 5' UTR model, our original method even outperformed the modified one, with statistically lower MAE and RMSE and a higher adjusted R-squared. Therefore, we believe our current approach remains appropriate.

**Author response image 4. sa4fig4:** 

"Overall, motifs that are at least two nucleotides long proved critical for RNA stability, supporting the sequence specificity of the decay process." Unclear why this supports the "sequence specificity"

No monomers were selected as an explanatory factor. On the contrary, specific sequence combinations and order are important for the regulation. These findings suggest sequence-specific recognition for the decay process.

Fig3: The same features were used in both cell lines? If yes: Since they were selected for their highest correlation with half-life, how was a common set chosen? If no: problematic to compare.

Thank you for your question regarding feature selection across cell lines. Initially, the features were collected uniformly for both cell lines. However, subsequent feature selection steps were cell-type specific, focusing on identifying features with the greatest impact on RNA half-life in each context. This approach allows us to still compare model performance and discuss the similarities and differences in selected features across cell types. By maintaining a consistent starting point, we ensure that any observed differences reflect cell-specific regulatory dynamics.

uORFs were not used as features?

Thank you for pointing this out. At the beginning of our study, we investigated the impact of Kozak sequence strength (categorized as weak, moderate, strong, or optimal) on RNA half-life. However, we found that this feature performed poorly in predicting RNA stability, and as a result, we decided not to include upstream open reading frames (uORFs) or Kozak sequences in our subsequent analyses.

Experimental reproducibility: Only correlations between replicates for the same time point is shown, but no comparison between time points or between decay rates. How reproducible were the paired differences between mut/wt?

The decay rate was calculated by modeling the slope of a linear regression of all time points. Therefore, there is only one decay rate associated with a genotype. To rule out inconsistent data, we excluded any regression with a mean square error greater than 1, as this indicates a poor fit of the data points.

Fig 7C/p17: This does not establish a "causal relationship" as the authors claim.

We agree with the reviewer’s suggestion. We have modified the text on p.17 to “to establish a correlation between UTR variants and health outcomes,…..”

In the discussion, the authors claim that TA-diNTs are not only an opposite of the GC percentage and base this on Fig 5A.

Fig 5A: The range of TA-diNTs is naturally much higher in the low GC group. To make the high and low GC content comparable (as the authors aim to do), the correlation should be assessed for the same range of TA dint in both cases.

To address this concern more rigorously, we performed a stratified analysis based on UA-diNT rate. As shown in our Fig. S7C, even after stratifying by UA- dinucleotide ratio (upper panel high UA- dinucleotide ratio / lower panel low UA- dinucleotide ratio), we still observe that the destabilizing effect of UA is stronger in the low GC content group.

Supplemental Figure S7. Interplay of GC content and TA dinucleotide on stability regulation, related to Figure 5. (C) Stratifications of both TA dinucleotide ratio and GC content showed that the destabilizing effect of TA dinucleotide is the most prominent under conditions of low TA dinucleotide ratio and low GC content. The same trend was observed for 5’ UTR (left) and 3’ UTR (right).

The injection of in vitro transcribed and polyA/capped RNA certainly has advantages over other methods, but delivering naked mRNA without nuclear history might also lead to artifacts. The caveats of the approach should be discussed more extensively.

We appreciate the suggestion and have hence added the following in the Discussion (p.18): “However, while our approach effectively assesses the stability of synthesized RNA in human cells, it may not fully capture the decay dynamics of nuclear-synthesized RNA, which can be influenced by endogenous modifications and trans-acting RNA binding factors.”

"We unexpectedly identified many crucial regulatory features in 5' UTRs." Why was this unexpected?

We initially thought the 3’ UTR would play a major role in stability regulation. To avoid confusion, we have removed the word ‘unexpected’ from the text (p. 20): "We identified many crucial regulatory features in 5' UTRs."

"...a massively parallel reporter assay in which coding regions and human 5'/3' UTRs with diseaserelevant mutations were generated in vitro and then directly transfected into human cell lines to assess their decay patterns by next‐generation sequencing": also coding regions?

Thanks for the question. Indeed, the coding region was not synthesized together with the UTR library. Therefore, we modified the text of p. 6 to “…we developed a massively parallel reporter assay in which human 5’/3’ UTRs with disease-relevant mutations were generated in vitro, ligated with the enhanced green fluorescence protein (EGFP) coding region, and then directly transfected into human cell lines to assess their decay patterns by next-generation sequencing.”

**Reviewer #2 (Recommendations For The Authors):**
Nomenclature: When discussing RNA sequences, "U" should be used in place of "T" (e.g., "UA dinucleotide").

We have replaced the RNA sequence “T” with “U” of the text and figures.

Abstract: "We examined the RNA degradation patterns mediated by the UTR library in multiple cell lines" - It would be clearer to state that two cell lines (rather than multiple) were used.

We appreciate the suggestion. We have modified the abstract as suggested: “We examined the RNA degradation patterns mediated by the UTR library in two cell lines…"

The manuscript refers to "wild-type (WT) and mutant (mt) alleles." (p. 7 and elsewhere). It would be better to use "reference" instead of "wild type" given that these are human populations.

We appreciate the suggestion. All instances of ‘wild-type’ or ‘WT’ in the text and figures have been replaced with ‘reference’ or ‘ref’.

In the introduction, it is stated that traditional MPRAs "cannot differentiate the effect of the UTRs on transcription, stability and, in some cases, even protein production, greatly limiting scientific interpretation." This is confusing, since these assays can and have been used in association with both RNA decay measurements and measurements of reporter protein levels that allow assessment of effects on stability and protein production (including in the cited references).

We reason that the RNA steady-state level (e.g., sequencing the overall RNA normalized to DNA) or protein steady-state level (e.g., detecting the fluorescence signal) does not precisely reveal the decay kinetics of the RNA. Steady-state level is a result of production and decay, both of which UTRs contribute to. Similarly, the protein level is not a perfect estimate of the RNA decay.

To clarify, we have modified the introduction (p. 5) to “Nevertheless, because the steady-state level is a result of production and decay, these approaches cannot differentiate the effect of the UTRs on transcription, stability and, in some cases, even protein production, greatly limiting scientific interpretation.”

Adding raw and normalized read count data from individual experiments (e.g., to Table S1) would make it more likely for others to use this dataset to address additional questions.

All raw and processed sequencing data generated in this study have been submitted to the NCBI Gene Expression Omnibus (GEO; https://www.ncbi.nlm.nih.gov/geo/) under accession number GSE217518 (reviewer token snspaakujtsdpcv).

The manuscript would benefit from further clarification about model selection. Additional details regarding how the features were clustered, and the actual clusters themselves should be included.It should be discussed why Lasso was chosen vs Ridge or Elastic Net, in the context of handling multicollinearity. Often, data is subsetted for training and validation, and model performance metrics are presented.

Thank you for pointing out the need for further clarification on model selection. The features were clustered using single-linkage hierarchical clustering with the distance metric defined as one minus the absolute value of the Spearman correlation coefficient (this information has been added to the manuscript on p. 28: “…to avoid multicollinearity caused by similar features that perturb feature selection, all features were clustered using single-linkage hierarchical clustering with the distance metric defined as one minus the absolute value of the Spearman correlation coefficient.”). The resulting feature clusters are available in Supplemental Table S3.

Regarding model selection, we chose LASSO over ridge and elastic net primarily for feature selection, as ridge does not perform feature selection. Elastic net is essentially a hybrid of ridge regression and LASSO regularization, but we opted for LASSO for its simplicity and effectiveness in selecting a sparse set of important features.

We also performed a 2:1 training and testing set analysis and have included these details in the manuscript. Model performance metrics, including correlation coefficient between observed and predicted values in the testing set, mean absolute error (MAE), root mean squared error (RMSE), mean absolute percentage error (MAPE), and R-squared, are provided in new Supplemental Table S4.

Recommend reviewing and correcting verb tenses in the methods section.

We appreciate the reviewer’s suggestion. We have corrected verb tenses in the methods section, which includes “The UTRs were defined by NCBI RefSeq and ENCODE V27. (p.21)”, “The variant was placed in the middle of the sequence….(p.22)”, and “eCLIP signals with value < 1 or p value > 0.05 were removed. (p.26)”

Please add information about which cell type(s) are being used in each of the figure legends (e.g., in Figs. 2B and 5).

We appreciate the reviewer’s suggestion. We have added the cell type information in the figure legends: “Figure 2…. (B) The ten most influential AREs in terms of RNA stability in SH-SY5Y cells.” And “Figure 5…..(A) MPRA data of SH-SY5Y cells stratified according to the GC content (GC%) of UTRs.”

Recommend review of axis labels and consistency in formatting the log(half-lives) and including the base of the log and the time unit (minutes). Even better, converting axis labels from log minutes to minutes would make this easier to understand.

Thank you for the suggestion regarding axis labels and consistency. We have unified the half-life label to ‘ln t_1/2_ (min)’ in all figures. We chose not to convert the axis from logarithmic minutes to minutes because the original scale is highly skewed, which would hinder clear data visualization.

The discussion refers to Figure 1D but Figure 1 only has A-C

Thank you for pointing out this mistake. ‘Fig. 1D’ has been changed to ‘Fig. 1B’ in the text (p. 7 and p. 20).

The analyses in Fig. 2 are interpreted as demonstrating that AREs destabilize RNAs. These analyses are examining associations, so it would be more appropriate to say that AREs are associated with destabilization (since it is formally possible that other sequences that are present in these UTR fragment cause destabilization). A similar issue arises on p. 10: "TA dinucleotides alone can negatively regulate RNA stability, with a Pearson's correlation coefficient of ‐0.287 for 5' UTRs and ‐0.377 for 3' UTRs (Fig. 4A,C)." This is an association and does not establish causation. Again on p. 17: "We identified several SNPs in UTRs that induce aberrant RNA expression and/or protein expression (Supplemental Table S7)." These may be causal but may simply be in LD with other variants that are causal.

We agree that the association observed is not proven to be causal. Therefore, we modified the text as suggested:

“AUUUA/AUUA-containing AREs are associated with RNA destabilization.” (p. 8)

“UA dinucleotides alone present a negative correlation with RNA stability, with a Pearson’s correlation coefficient of -0.287 for 5’ UTRs and -0.377 for 3’ UTRs.” (p.10)

“We identified several SNPs in UTRs that correlated with aberrant RNA expression and/or protein expression.” (p. 17)

Figure 4C is important in that it examines whether variant sequences that differ in a manner that changes the number of dinucleotide repeats affect stability. Please show the number (not just the percentage) of sequences in each category.

Thank you for your insightful comment. We believe the figure you referred to is Figure 4E. We have updated the figure to include the number of sequences in each category.

Figure 6A and B: The horizontal axes appear to be misaligned since the dotted vertical lines do not cross at 0. ?

The dotted vertical lines represent the genomic background of the UA-diNT ratio. To clarify it, we have modified the legend to: “Figure 6……(A) The top ten biological processes for which the 5’ UTR UA-dinucleotide ratio most significantly deviated from the genomic background (dashed line).”

It may be helpful to state what the dashed and solid lines represent on Figure 6 E/F. Please correct spelling of "Biological" in 6E.

As per the reviewer’s suggestions, we have modified the legend of Figure 6 to: “………..(E) Biological processes for RNAs in which the UA-dinucleotide ratios of both 5’ and 3’ UTRs are significantly different from the genomic background (dashed lines). (F) Molecular functions for RNAs in which the UA-dinucleotide ratios of both 5’ and 3’ UTRs are significantly different from the genomic background (dashed lines). The thin solid lines represent the standard deviation of the UAdinucleotide ratio within the gene group.”

In addition, the spelling of “Biological” in Fig. 6E has been corrected.

**Reviewer #3 (Recommendations For The Authors):**
I have 3 points that I think could improve science and its presentation within the manuscript.(1) Most importantly, how well do LASSO regression models predict the stability of native transcripts? Such analysis can also be useful for comparison between two different cell-types. How well does the regression model learned (on reporters) within one cell-type predict mRNA stability (of reporters and native genes) in this cell-type and in the other cell-type? Similarly, models can also help to analyze the effects of 5'UTR and 3'UTR sequences on mRNA stability. In particular, how well does the regression model of each separate regulatory sequence (3'UTR or 5'UTR) is able to predict the stability of native genes in the cell? Can the predictions be improved by combining both 3'UTR and 5'UTR sequence features within the regression models?

The decay model for native transcripts has been established in prior research (doi.org/10.1186/s13059-022-02811-x; doi.org/10.1186/s12915-021-00949-x), which indicates that exon junction density and transcript length are the primary determinants of RNA stability. Based on these findings, we designed the MPRA with fixed length and without splicing to focus on the contribution of primary sequences. We validated the destabilizing effect of UA dinucleotide on endogenous RNAs (Fig. 4G and Supplemental Fig. S5F) but do not recommend using our model to fully explain or predict the stability of native transcripts.

To assess the model's cross-cell type predictive performance for RNA half-life, we employed the Regression Error Characteristic (REC) curve (Bi & Bennett, 2003). Similar to the receiver operating characteristic (ROC) curve, the REC curve illustrates the trade-off between error tolerance and accuracy, with better performance indicated by curves trending toward the upper left. We also computed the Area Over the Curve (AOC) as a performance metric, where lower values indicate better predictive ability. From Author response image 5, the REC curves reveal that cross-cell type prediction performance is suboptimal. The y-axis represents prediction accuracy, while the x-axis denotes error tolerance for the natural logarithm of RNA half-life (ln(*t1/2*), in minutes).

**Author response image 5. sa4fig5:** 

In response to the suggestion of combining 5' and 3' UTR sequence features in the regression model, we believe this approach may not be ideal. As shown in Figure S1D, the distribution of RNA half-lives between 5' and 3' UTRs is significantly different, reflecting their distinct regulatory roles. Additionally, the base composition differs, with 5' UTRs having a higher GC content compared to 3' UTRs. Combining these datasets would likely make the origin of the sequence (5' or 3' UTR) the most predictive feature, thereby reducing the model's interpretability. Furthermore, our MPRA results, derived from separate 5’ or 3’ UTR library, do not support a combined model, further suggesting this approach may not be suitable with our data.

The conclusions regarding genetic variants are interesting, yet the main strength of the work involves identifying general sequence features that affect mRNA stability rather than specific variants. I wonder if the authors have considered to shift the focus of the motivation part to reflect that?

We appreciated the reviewer’s suggestion. We have revised the abstract and introductions to emphasize the general UTR regulation. Here is the revised abstract:

UTRs contain crucial regulatory elements for RNA stability, translation and localization, so their integrity is indispensable for gene expression. Approximately 3.7% of genetic variants associated with diseases occur in UTRs, yet a comprehensive understanding of UTR variant functions remains limited due to inefficient experimental and computational assessment methods. To systematically evaluate the effects of UTR variants on RNA stability, we established a massively parallel reporter assay on 6,555 UTR variants reported in human disease databases. We examined the RNA degradation patterns mediated by the UTR library in two cell lines, and then applied LASSO regression to model the influential regulators of RNA stability. We found that UA dinucleotides and UA-rich motifs are the most prominent destabilizing element. Gain of UA dinucleotide outlined mutant UTRs with reduced stability. Studies on endogenous transcripts indicate that high UA-dinucleotide ratios in UTRs promote RNA degradation. Conversely, elevated GC content and protein binding on UA dinucleotides protect high-UA RNA from degradation. Further analysis reveals polarized roles of UA-dinucleotide-binding proteins in RNA protection and degradation. Furthermore, the UA-dinucleotide ratio of both UTRs is a common characteristic of genes in innate immune response pathways, implying a coordinated stability regulation through UTRs at the transcriptomic level. We also demonstrate that stability-altering UTRs are associated with changes in biobank-based health indices, underscoring the importance of precise UTR regulation for wellness. Our study highlights the importance of RNA stability regulation through UTR primary sequences, paving the way for further exploration of their implications in gene networks and precision medicine.

Plots presenting correlations (e.g., Figure 4A, 4C) are more informative when plotted as density plots (i.e., using colorscale to show density of the dots at each part of the plot).

We greatly appreciate the reviewer's insightful suggestion regarding the use of density plots for presenting correlations. We have modified Figures 4A and 4C in the revised manuscript to implement density plotting. The updated figures now utilize a colorscale that highlights areas of high and low data density.